# ARMOR: High-Performance Semi-Structured Pruning via Adaptive Matrix Factorization

**Lawrence Liu**[1] * **Alexander Liu**[2] **Mengdi Wang**[3] **Tuo Zhao**[4] **Lin F. Yang**[1]
[1]*University of California, Los Angeles*    [2]*Independent*    [3]*Princeton University*
[4]*Georgia Institute of Technology*

## Abstract

Large language models (LLMs) present significant deployment challenges due to their immense computational and memory requirements. While semi-structured pruning, particularly 2:4 sparsity, offers a path to practical hardware acceleration, existing methods often incur substantial performance degradation. To bridge this gap, we introduce **ARMOR**: (**A**daptive **R**epresentation with **M**atrix-fact**OR**ization), a novel one-shot post-training pruning algorithm. Instead of directly pruning weights, ARMOR factorizes each weight matrix into a 2:4 sparse core wrapped by two low-overhead, block diagonal matrices. These wrappers act as efficient pre- and post-transformation error correctors, offering greater flexibility to preserve model quality compared to conventional 2:4 pruning techniques. The sparse core and block diagonal wrappers are chosen through a block coordinate descent algorithm that minimizes a layer-wise proxy loss. We prove this optimization is guaranteed to converge to a solution with a proxy loss less than or equal to state-of-the-art pruning algorithms. Experiments on Llama (Touvron et al., 2023; Dubey et al., 2024) and Qwen (Yang et al., 2025) model families demonstrate that ARMOR consistently and significantly outperforms state-of-the-art 2:4 pruning methods across a wide range of downstream tasks and perplexity evaluations, and generalizes to provide improvements for general N:M patterns and unstructured sparsity. ARMOR achieves this superior performance while retaining the inference speedups and substantial memory usage reductions of 2:4 pruning, establishing a more effective trade-off between model compression and task accuracy.

## 1 Introduction

Large Language Models (LLMs) have demonstrated remarkable capabilities (Park et al., 2023; Huang & Yang, 2025), yet their immense computational and memory requirements pose significant barriers to practical deployment. As a result, techniques for reducing model sizes and computational costs while retaining performance are of significant interest for the research community. A particular area of focus are one-shot post training compression techniques, a highly efficient model compression regime where already trained models are compressed in a single pass without iterative fine-tuning. Pruning, the removal of model parameters is a particularly compelling avenue of compression as it offers a direct path to inference acceleration via dedicated hardware support for specific sparsity patterns (Kwon et al., 2022), and its benefits can be compounded with orthogonal methods like quantization (Frantar et al., 2022; Lin et al., 2024; Li et al., 2025a). However, a critical trade-off plagues existing pruning techniques. Methods capable of delivering tangible inference acceleration do so by sacrificing significant model accuracy, while the most accurate techniques offer largely theoretical speedups. Bridging this gap is the central focus of our work.

Pruning algorithms can be broadly divided into three categories: structured, unstructured, and semi-structured pruning. Structured pruning removes entire weight structures, such as rows or columns of weight matrices (Ashkboos et al., 2024), attention heads (Ma et al., 2023), or even full layers (Men et al., 2024). This coarse-grained approach is highly compatible with existing hardware and software, as it results in smaller, dense matrices that can be processed efficiently by standard libraries,

---

*Correspondence to `lawrencerliu@ucla.edu`

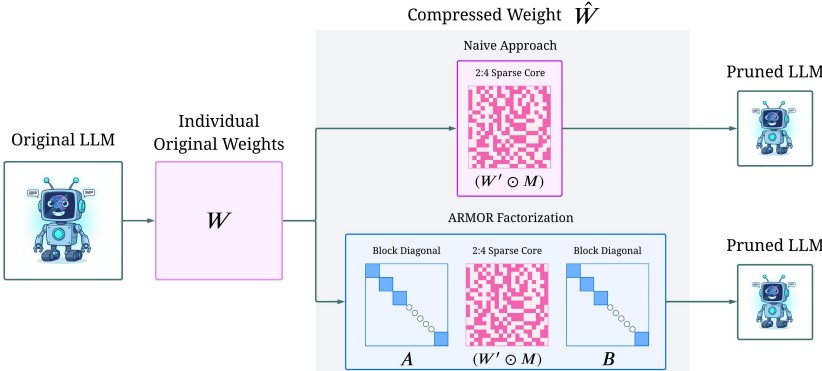

Figure 1: Illustration of proposed ARMOR factorization. For a given LLM, each weight matrix $W$ is pruned individually. Instead of naively pruning the weight matrix, ARMOR wraps the sparse core with a pair of block diagonal matrices and uses a unique optimization algorithm to find the optimal structured pruning mask. $M \in \{0, 1\}^{d_{\text{out}} \times d_{\text{in}}}$ represents the 2:4 binary mask.

leading to direct improvements in inference speed (Ma et al., 2023). However, this rigidity comes at a cost; by removing large, contiguous blocks, structured pruning can lead to a significant degradation in model accuracy and often has a lower limit on the achievable sparsity before performance collapses (Ashkboos et al., 2024).

At the other end of the spectrum, unstructured pruning offers the highest flexibility by removing individual weights from any part of the model. Leading unstructured pruning algorithms have shown that it is possible to remove up to 50% of the weights from an LLM with minimal loss in performance (Frantar & Alistarh, 2023). However, the resulting irregular, sparse matrices disrupt the parallel processing capabilities of modern hardware like GPUs, which are optimized for dense matrix operations. Thus it is difficult to translate the theoretical model size reductions into a practical inference speedup (Xia et al., 2023).

To bridge the gap between hardware efficiency and model performance, semi-structured pruning has emerged as a compelling compromise. A popular variant is N:M sparsity, which enforces a regular pattern by ensuring that within any contiguous block of M weights, only N weights are non-zero. This regularity is key, as it can be directly accelerated by specialized hardware. For instance, NVIDIA's Ampere and subsequent GPU architectures provide native support for 2:4 semi-structured sparsity, which can theoretically double the throughput of matrix operations (Hu et al., 2024; Mishra et al., 2021). However, the constraint of a fixed pruning pattern within each small block restricts the algorithm's ability to retain the most critical weights, leading to a significantly increased drop in performance. For example, applying a state-of-the-art 2:4 pruning method to Llama-7B increases Wikitext2 perplexity by nearly 59% over its 50% unstructured counterpart, creating an undesirable choice between theoretical efficiency and practical accuracy (Sun et al., 2024).

In this work, we seek to close this performance gap by introducing **ARMOR**: (**A**daptive **R**epresentation with **M**atrix-fact**OR**ization), an theoretically grounded one-shot model pruning algorithm. We reframe semi-structured pruning as a matrix factorization problem. Instead of directly pruning weights, our key insight is to factor each weight matrix into a constrained sparse core that adheres to the 2:4 hardware pattern, pre- and post-multiplied by lightweight block diagonal matrices, this factorization is illustrated in Figure 1. These block diagonal matrices, which are highly parameter-efficient due to their sparse structure (containing only $O(N)$ parameters compared to $O(N^2)$ for a dense matrix), can be multiplied with activations efficiently on modern hardware. They act as learned, low-overhead linear transformations that rotate the activation and weight spaces into a basis where the 2:4 pruning constraint is less lossy, providing enhanced flexibility and preserving model quality more effectively than naive 2:4 pruning algorithms.

Through extensive experiments on Llama (Dubey et al., 2024; Touvron et al., 2023) and Qwen (Yang et al., 2025) family models, we show that ARMOR consistently and significantly outperforms existing 2:4 pruning methods on both perplexity and downstream tasks. For example, on Llama-2-

13B, ARMOR reduces the perplexity gap between the 2:4 pruned model and the dense original by almost 50%. We show that these accuracy gains are achieved while preserving the practical inference speedups and memory reduction inherent to native 2:4 sparsity, and that ARMOR generalizes to general N:M and unstructured sparsity with consisted improvements over SOTA. Our work suggests that rethinking the fundamental representation of weights, rather than simply removing them, is a promising direction for future hardware-software co-design in efficient deep learning.

## 1.1 RELATED WORK

Existing one-shot unstructured/semi-structured pruning methods largely formulate the compressed weight matrix $\hat{W}$ as the element-wise product of a dense matrix $W'$ and a binary mask $M$, where $\hat{W} = W' \odot M$. Research within this paradigm focuses on two primary challenges: identifying an optimal mask $M$ (importance scoring) and updating the unpruned weights in $W'$ to compensate for the removed connections. The simplest approaches are weight-update-free methods like Wanda, which fix the unpruned weights to their original values ($W' = W$) and focus on finding a good mask M through element-wise metrics (Sun et al., 2024; Liu et al., 2025; Liu et al.; Dong et al., 2024; Das et al., 2023; Zhang et al., 2024). In contrast, more complex weight-update methods such as SparseGPT (Frantar & Alistarh, 2023) aim for higher accuracy by iteratively pruning weights while simultaneously adjusting the remaining ones to minimize a proxy reconstruction loss, often based on a Hessian sketch. While effective, this introduces significant computational overhead, such as the costly inversion of the Hessian sketch matrix. Both methods suffer significantly increased performance loss when applied to 2:4 semi-structured pruning compared with their unstructured counterparts.

This gap has led to alternative formulations beyond elementwise masking. These included sparse matrix factorization approaches such as DSF (Boža & Macko, 2024), decomposition based works like WRP, and hybrid low-rank structures such as LoSparse (Li et al., 2023) and Targeted Low Rank Refinement (Shen et al., 2025). However such approaches suffer from their own challenges, DSF and outlier-adaptive schemes like OWL (Yin et al., 2023) are incompatible with accelerated 2:4 kernels. Similarly, WRP's performance is bottlenecked by its reliance on a highly unstructured sparse matrix component, for which no efficient matrix-multiplication kernels exist. Directly learnable approaches such as LoSparse and MaskLLM (Fang et al., 2024) typically rely on expensive iterative retraining, for instance MaskLLM uses $\sim 4000\times$ more data and $\sim 80\times$ more compute than ARMOR. Finally, rotation-based methods such as RotPruner (Chen & Wang, 2025) and DenoiseRotator (Gu et al., 2025) introduce fixed overheads. Unlike ARMOR 's tunable block-diagonal wrappers, these rotations cannot be easily adjusted to balance accuracy and latency, leaving a gap between high compression performance and tangible speedups.

## 2 PROBLEM STATEMENT

To remain computationally tractable, one-shot compression pruning methods regularly adopt a layer-by-layer framework. For a given linear layer with weight matrix $W \in \mathbb{R}^{d_{\text{out}} \times d_{\text{in}}}$, the objective is to find a compressed representation $\hat{W}$ that minimizes a data-aware proxy loss, $\mathcal{L}_{W,X}(\hat{W})$. This loss quantifies the approximation error using a small calibration dataset $X^T \in \mathbb{R}^{n \times d_{\text{in}}}$. The exact formulation of this proxy loss is a key design choice and varies across different pruning algorithms, we define our proxy loss in Section 3.2.

We constrain $\hat{W}$ to leverage a 2:4 semi-structured sparse component to guarantee hardware acceleration. This requires that the underlying binary mask $M \in \{0,1\}^{d_{\text{out}} \times d_{\text{in}}}$ defining this component's pattern has exactly two non-zero entries in every group of four consecutive columns per row. Formally we express this constraint as $\left\|M_{i,[k]}\right\|_0 = 2, \quad \forall i \in [d_{out}], k \in [d_{in}/4]$ where we adopt $M_{i,[k]} = M_{i,4(k-1)+1:4k}$ as shorthand. The general layer-wise optimization problem is therefore:

$$\min_{\text{params of } \hat{W}} \mathcal{L}_{W,X}(\hat{W}) \quad \text{s.t.} \quad \left\|M_{i,[k]}\right\|_0 = 2, \quad \forall i \in [d_{out}], k \in [d_{in}/4]$$

Regardless of optimization algorithm, this layer-wise approach is inherently one-shot because the compression is completed in a single pass over the network's layers without any global retraining.

## 3  METHODS

In this section we introduce the ARMOR pruning process. Section 3.1 introduces the ARMOR factorization and notation. In Section 3.2 we discuss the proxy loss optimization objective and initialization. In Section 3.3 we introduce the ARMOR optimization algorithm to optimize the ARMOR factorization. This algorithm consists of two alternating steps, the continuous parameter update step, Section 3.3.1, and the sparse core update step, Section 3.3.2. Finally we introduce the main theoretical result in 3.4.

### 3.1  ARMOR MATRIX FACTORIZATION

For each layer in an LLM, let the original weight matrix be $W \in \mathbb{R}^{d_{\text{out}} \times d_{\text{in}}}$. Our approach seeks to find a compressed matrix, $\hat{W} \in \mathbb{R}^{d_{\text{out}} \times d_{\text{in}}}$, with the following factorization:

$$\hat{W}(A, B, W', M) := A \cdot (W' \odot M) \cdot B, \qquad (1)$$

where the goal is to optimize for the parameters $A, B, W'$, and $M$. Specifically, $W' \in \mathbb{R}^{d_{\text{out}} \times d_{\text{in}}}$ is a dense matrix representing transformed weights, and $M \in \{0,1\}^{d_{\text{out}} \times d_{\text{in}}}$ is a binary mask that imposes sparsity through element-wise multiplication ($\odot$). The matrices $A \in \mathbb{R}^{d_{\text{out}} \times d_{\text{out}}}$ and $B \in \mathbb{R}^{d_{\text{in}} \times d_{\text{in}}}$ are block-diagonal. The block size, $d_{\text{block}}$, is a chosen hyperparameter, selected such that it divides both $d_{\text{out}}$ and $d_{\text{in}}$. We refer to the set of all learnable parameters as $\theta = (A, B, W', M)$.

The key to ARMOR is the diagonal matrix wrappers, $A$ and $B$, that surround the sparse "core" $W' \odot M$. Compared to the naive approach of directly pruning or diagonal only wrappers, these block diagonal matrix wrappers offer additional flexibility, while having low overhead and existing implementations on hardware for storage and inference. We can store $A$ and $B$ as tensors of size $(d_{\text{out}}/d_{\text{block}}) \times d_{\text{block}} \times d_{\text{block}}$ and $(d_{\text{in}}/d_{\text{block}}) \times d_{\text{block}} \times d_{\text{block}}$ respectively. Matrix multiplication at inference time can be performed as batched matrix multiplication. As a result the overhead of storing $A$ and $B$ and performing inference grows with $\mathcal{O}((d_{\text{out}} + d_{\text{in}})d_{\text{block}})$, which is sublinear to the number of original parameters in the layer, $d_{\text{out}}d_{\text{in}}$.

**Notation**  We denote the individual blocks of $A$ and $B$ as $A^{(i)} \in \mathbb{R}^{d_{\text{block}} \times d_{\text{block}}}$ and $B^{(j)} \in \mathbb{R}^{d_{\text{block}} \times d_{\text{block}}}$, ie: $A = \text{diag}\left(A^{(1)}, A^{(2)}, ..., A^{(d_{\text{out}}/d_{\text{block}})}\right)$ and likewise for $B$. More generally for any matrix $C \in \mathbb{R}^{d_{\text{out}} \times d_{\text{in}}}$ we denote the $d_{\text{block}} \times d_{\text{block}}$ matrix blocks as $C^{(i,j)}$, ie $C^{(i,j)} = C_{(i-1)d_{\text{block}}+1:id_{\text{block}},(j-1)d_{\text{block}}+1:jd_{\text{block}}} \quad \forall i \in [d_{\text{out}}/d_{\text{block}}], \quad j \in [d_{\text{in}}/d_{\text{block}}]$. A detailed illustration of this notation can be found in appendix A.

### 3.2  OPTIMIZATION OBJECTIVE

We optimize $\hat{W}$ by minimizing the NoWag layerwise proxy loss (Liu et al., 2025).

$$\mathcal{L}_{W,X}(\theta) = \|\bar{W} - \hat{W}\|^2_{F,\text{diag}(XX^T)} := \sum_i \sum_j (\bar{W}_{ij} - \hat{W}_{ij})^2 \|X_j\|_2^2, \qquad (2)$$

where $\bar{W} \in \mathbb{R}^{d_{\text{out}} \times d_{\text{in}}}$ is the row and column normalized version of $W$:

$$\bar{W}_{ij} = \frac{1}{r_i^{(2)}} \left( \frac{W_{ij}}{r_j^{(1)}} \right), \quad r_j^{(1)} = \sqrt{\sum_{i=1}^{d_{\text{out}}} W_{ij}^2}, \quad \forall j \in [d_{\text{in}}], \quad r_i^{(2)} = \sqrt{\sum_{j=1}^{d_{\text{in}}} \left( \frac{W_{ij}}{r_j^{(1)}} \right)^2}, \quad \forall i \in [d_{\text{out}}].$$

We adopt the NoWag objective function due to its data-aware nature and decomposable structure. The $\text{diag}\left(XX^T\right)$ term weights the squared Frobenius norm by the magnitude of the corresponding input activations, which focuses the optimization on preserving weights that are most influential for the given calibration data. Furthermore, compared to Hessian sketch based methods, this objective function requires less calibration data.

A key advantage of this objective is that the loss can be decomposed into independent element-wise subproblems. While ARMOR factorization's wrapper matrices prevent a direct element-wise optimization, this structure is still critical as it allows for the problem to be broken down into independent block-level subproblems, which is a cornerstone of our greedy sparse core optimization strategy. After optimizing, we scale $\hat{W}$ back by denormalizing, which is performed by pre-scaling the rows and columns of $A$ and $B$ by $r^{(1)}$ and $r^{(2)}$ respectively before inference.

### 3.3 OPTIMIZATION ALGORITHM

---

**Algorithm 1** ARMOR Optimization Algorithm

---

**Require:** Original weight $W$, calibration data $X$, tolerance $\epsilon$.
**Ensure:** Optimized factorization parameters $\theta = \{A, B, W', M\}$.
1:                                                            $\triangleright$ Initialization
2:   $\overline{W} \leftarrow \text{Normalize}(W)$                      $\triangleright$ Using row and column normalization from NoWag
3:   $(\theta)_0 \leftarrow \text{Initialize}(\overline{W}, X)$                           $\triangleright$ Initialize according to Eq 3
4:   **for** $t = 1, 2, 3, ..., n_{\text{iters}}$ **do**
5:       $(A)_t, (B)_t, (W')_{t|\text{cont}} \leftarrow \text{ContinuousUpdate}((A)_{t-1}, (B)_{t-1}, (W')_{t-1}, (M)_{t-1}, \overline{W}, X)$
6:       $(W')_t, (M)_t \leftarrow \text{SparseCoreUpdate}((A)_t, (B)_t, (W')_{t|\text{cont}}, (M)_{t-1}, \overline{W}, X)$
7:   **end for**
8:   **return** $A_t, B_t, W'_t, M_t$

---

To optimize the ARMOR factorization, we utilize a block coordinate descent (Wright, 2015) algorithm that alternates between updating the continuous parameters $A, B, W'$ and the sparse core $W' \odot M$. These alternating updates are performed for $n_{\text{iters}}$ iterations. The overall algorithm is outlined in Algorithm 1. A detailed technical description of the optimization algorithm can be found in appendix B.

**Initialization**    We initialize our factorization $(\theta)_0 = ((A)_0, (B)_0, (W')_0, (M)_0)$ as:

$$(A)_0 = I, \quad (B)_0 = I, \quad (W')_0 = \bar{W},$$
$$(M_{ij})_0 = \mathbb{I}\left(\mathcal{I}_{i,j} \in \text{top}_2\left(\mathcal{I}_{i,[c]}\right)\right) \forall i \in [d_{\text{out}}], j \in [d_{\text{in}}], \quad (3)$$

where $c = \lceil j/4 \rceil$, $\mathcal{I}_{i,j} = \bar{W}_{i,j}^2 \|X_j\|_2^2$, and $I$ denotes the identity matrix of appropriate size. This initialization is chosen as it is the optimal solution to equation 2 if $\hat{W}$ only consisted of the naive sparse core, and is equivalent to the pruning result of NoWag-P pruning algorithm (Liu et al., 2025).

#### 3.3.1 UPDATING OF CONTINUOUS PARAMETERS

This step uses sequential gradient descent to update $A$, $B$, and $W'$ sequentially, with the learning rate determined via local $\beta$-smoothness. Algorithm 2 depicts a single step in detail. In practice, we replace these sequential steps with a joint Adam (Kingma & Ba, 2014) optimization that updates $A$, $B$, and $W'$ simultaneously, a choice driven primarily by efficiency. This approach requires only one forward/backward pass per iteration and eliminates the need to recalculate local $\beta$-smoothness at each step. While Adam introduces minor additional memory overhead, this is negligible since we optimize each layer independently. We present the sequential gradient descent version here to provide strict theoretical guarantees of convergence. In practice, however, joint Adam optimization yields no significant differences compared to sequential gradient descent.

#### 3.3.2 UPDATING THE SPARSE CORE

This step updates the sparse core $W' \odot M$ to reduce the proxy loss. To avoid the exponential large search space, we adopt a greedy approach, where we select and update a fraction of the elements of the sparse core to reduce the proxy loss. An illustration of the algorithm is depicted in Figure 2 and a mathematical description can be found in Appendix B.1. Below we elaborate the technical novelty and efficiency of the procedure.

**Leveraging the 2:4 Pattern**    If we freeze all of the sparse core beyond a single consecutive sparse group, finding the optimal values for this sparse group can be performed as follows. Since there are only $\binom{4}{2} = 6$ possible mask choices for a group, it is computationally tractable to consider each possible mask. For a possible mask choice, finding the proxy loss minimizing values for the 2 nonzero elements of the group is a least squares problem. Thus we can solve the least squares for all 6 possible mask choices and select the one with the smallest minimal proxy loss.

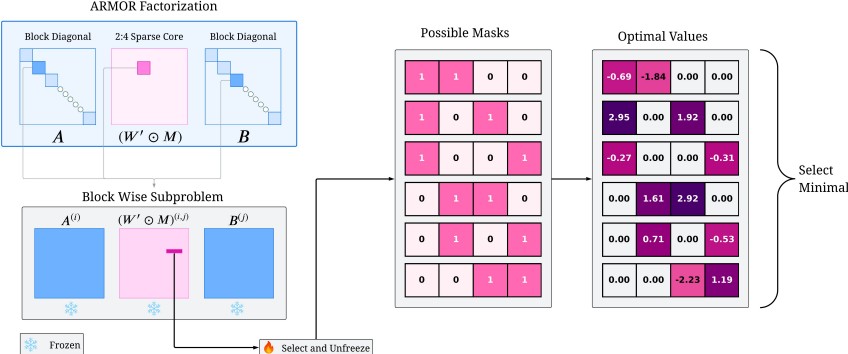

Figure 2: An illustration of the sparse core update step of the ARMOR optimization algorithm

**Leveraging the Elementwise Property of the Proxy Loss**   Updating a single 4 element sparse group usually has minimal impact on the layerwise proxy loss, since 4 elements account for less than $0.2 \times 10^{-6}$ of the parameters of a typical LLM layer. To enable updates of more than 4 elements at a time, we leverage the element wise property of the proxy loss. As discussed previously, this allows us to break the optimization problem into broken independent block-level subproblems:

$$
\mathcal{L}_{W,X}(\theta) = \sum_{i=1} \sum_{j=1} \ell_X^{(i,j)}(\theta^{(i,j)})
$$
$$
= \sum_{i=1} \sum_{j=1} \left\| \bar{W}^{(i,j)} - A^{(i)} \left( W'^{(i,j)} \odot M^{(i,j)} \right) B^{(j)} \right\|_{F,\text{diag}(XX^T)^{(j)}}^2 , \quad (4)
$$

where $i \in [d_{\text{out}}/d_{\text{block}}], \quad j \in [d_{\text{in}}/d_{\text{block}}]$. Thus we can optimize each block, $\left( W'^{(i,j)} \odot M^{(i,j)} \right)$, independently through the greedy least squares process outlined above. This allows for parallel updates of $(d_{\text{in}} d_{\text{out}})/d_{\text{block}}^2$ groups at a time, which for a standard LLM translates to on the order of $10^3$ more elements updated at once.

**Selecting The Sparse Group**   We select the sparse groups randomly, with their selection probabilities weighted by their proxy loss gradient. For block $(i,j)$, the probability $p_{(i',k)}^{(i,j)}$ of selecting group $i', k$ is:

$$
p_{(i',k)}^{(i,j)} \propto \left\| \left( \nabla_{(W' \odot M)^{(i,j)}} \ell_X^{(i,j)} \right)_{i',[k]} \right\|_1 \quad \forall \quad i' \in [d_{\text{block}}], \quad k \in [d_{\text{block}}/4].
$$

Such a selection heuristic results in more focus on the important sparse groups through proxy loss gradient weighting. Additionally, the randomness helps prevent selecting the same group over and over again. Empirically we observer that this leads to faster and better convergence of the proxy loss, and thus better overall LLM performance retention. An ablation of this selection heuristic can be found in Appendix E.1.

## 3.4   THEORETICAL RESULTS

In this section, we establish the following theorem to show that our algorithm guarantees convergence

**Theorem 3.1.** *(Convergence of the ARMOR optimization algorithm).   The sequence* $\{\mathcal{L}_{W,X}((\theta)_t)\}_{t \geq 0}$ *converges and* $\mathcal{L}_{W,X}((\theta)_t) \leq \mathcal{L}_{W,X}((\theta)_0) \quad \forall \quad t > 0.$

The proof can be found in Appendix C. Since ARMOR factorization initialization is equivalent to NoWag-P, $\mathcal{L}_{W,X}((\theta)_0)$ is the proxy loss of NoWag-P. Thus ARMOR will perform at least equivalently to NoWag-P a SOTA pruning algorithm, in terms of proxy loss.

## 4 RESULTS

**Models and Experimental Setup** We demonstrate the efficacy of our pruning method on two contemporary model families: Qwen 2.5 (7B, 14B, 32B, and 72B) and Qwen 3 (8B and 14B) (Yang et al., 2025). Our investigation targets foundational base models of 7B parameters and larger. This focus on base models allows for a direct assessment of our algorithm's effect on the core knowledge acquired during pre-training, removing confounding variables introduced by instruction tuning or other post-training modifications. Furthermore, we concentrate on dense architectures, excluding Mixture-of-Experts (MoE) variants, as recent work suggests that MoE models benefit from specialized pruning strategies (Xie et al., 2024; Li et al., 2025b). For all pruning experiments, we configured the block size, $d_{\text{block}}$, to 128 and executed the proxy loss optimization for 20,000 iterations. Additional implementation details are provided in Appendix H.

**Evaluation:** To comprehensively assess performance degradation, we employed a two-pronged evaluation strategy. First, to measure practical performance on downstream tasks, we evaluated the pruned Qwen models on a suite of seven industry-standard benchmarks using the LM Eval Harness (Gao et al., 2024). These benchmarks cover a range of capabilities, including commonsense and complex reasoning, mathematical problem-solving, and world knowledge. A detailed description of each benchmark is available in Appendix G. Second, to ensure comparability with the broader model compression literature, which often relies on perplexity metrics, we conducted an additional set of experiments. For this, we pruned models from the Llama-2 (7B, 13B, and 70B) (Touvron et al., 2023) and Llama-3 (8B and 70B) (Dubey et al., 2024) families. We then evaluated their perplexity on the test split of Wikitext2 (Merity et al., 2016) and a subset of the C4 validation split (Dodge et al., 2021), following standard evaluation protocols in the field.

**Baselines** We compare ARMOR against 3 leading pruning methods, SparseGPT (Frantar & Alistarh, 2023), Wanda (Sun et al., 2024), NoWag-P (Liu et al., 2025). Using NoWag-P is of particular interest, since as discussed previously, ARMOR uses the same proxy loss and is initalized at NoWag-P. Therefore comparing ARMOR against NoWag-P servers as an ablation to evaluate the empirical effectiveness of the ARMOR factorization and optimization algorithm.

### 4.1 TASK BASED EVALUATIONS

Results of the task based evaluations on Qwen 2.5 (7B/14B/32B/72B) and Qwen 3 (8B/14B) models are shown in Tables 1 and 2 respectively. Across all 7 tasks and all models evaluated, ARMOR consistently and significantly outperforms the state-of-the-art pruning methods. For example, on GPQA with the Qwen 2.5-32B model, ARMOR achieves a score of 39.51, outperforming even the dense model's performance (38.84) and vastly exceeding the next best pruning method, SparseGPT, which scored a 30.36. The performance gains are especially pronounced in reasoning or domain expertise heavy tasks like GSM8K, BBH, GQPA, demonstrating that our factorization approach is more effective at preserving the complex capabilities of the model compared to simply removing weights.

### 4.2 PERPLEXITY BASED EVALUATIONS

Results of perplexity based evaluations on Llama-2 (7B/13B/70B) and Llama-3 (8B/70B) models are reported in Table 3. Closely reflecting the task based results, ARMOR consistently and significantly outperforms the state-of-the-art pruning methods in retaining lower perplexity, a key indicator of language modeling quality. For example, on Llama-2-13B evaluated on Wikitext2, ARMOR achieves a perplexity of 6.37. This is a dramatic improvement over the next-best baseline (NoWag-P at 8.28) and represents a reduction of nearly 50% in the perplexity gap relative to the original dense model. We observe similar substantial gains across all evaluated Llama models and datasets, reinforcing that the ARMOR factorization preserves model quality more effectively than existing 2:4 pruning techniques.

### 4.3 COMPARISONS WITH LEARNABLE BASELINES

We further compare ARMOR against recent learnable semi-structured pruning methods: RotPruner (Chen & Wang, 2025) and DenoiseRotator (Gu et al., 2025). For these specific comparisons, we

| Method | Sparsity | Task Accuracy (%) (↑) | | | | | | |
|---|---|---|---|---|---|---|---|---|
| | | MMLU | GSM8K | BBH | GPQA | ARC-C | Wino | Hella |
| Dense (2.5-7B) | 0 | 74.19 | 82.33 | 69.16 | 33.03 | 59.55 | 76.09 | 60.03 |
| Dense (2.5-14B) | 0 | 79.8 | 88.02 | 75.18 | 38.17 | 64.51 | 80.9 | 63.46 |
| Dense (2.5-32B) | 0 | 83.24 | 88.78 | 81.72 | 38.84 | 66.30 | 81.22 | 65.12 |
| Dense (2.5-72B) | 0 | 86.06 | 89.54 | 85.01 | 42.63 | 68.52 | 82.16 | 67.63 |
| SparseGPT (2.5-7B) | 2:4 | 56.91 | 36.69 | 46.31 | 29.69 | 43.43 | 68.35 | 47.04 |
| Wanda (2.5-7B) | 2:4 | 52.21 | 31.00 | 41.39 | 25.45 | 37.80 | 63.37 | 43.81 |
| NoWag-P (2.5-7B) | 2:4 | 53.51 | 28.28 | 39.98 | 27.23 | 39.16 | 64.01 | 44.33 |
| ARMOR (2.5-7B) | 2:4+4.95% | **65.56** | **53.28** | **55.11** | **31.47** | **48.63** | **70.96** | **51.67** |
| SparseGPT (2.5-14B) | 2:4 | 64.21 | 46.55 | 56.44 | 31.92 | 48.21 | 71.51 | 49.73 |
| Wanda (2.5-14B) | 2:4 | 59.98 | 49.36 | 54.51 | 29.91 | 46.16 | 69.38 | 48.24 |
| NoWag-P (2.5-14B) | 2:4 | 58.45 | 45.87 | 52.1 | 30.36 | 43.77 | 68.67 | 48.42 |
| ARMOR (2.5-14B ) | 2:4+4.17% | **70.55** | **67.17** | **67.24** | **33.48** | **53.75** | **74.82** | **55.18** |
| SparseGPT (2.5-32B) | 2:4 | 75.26 | 66.03 | 71.31 | 30.36 | 57.08 | 79.01 | 55.73 |
| Wanda (2.5-32B) | 2:4 | 75.45 | 72.93 | 70.17 | 35.27 | 55.72 | 77.82 | 55.71 |
| NoWag-P (2.5-32B) | 2:4 | 74.89 | 68.69 | 69.53 | 27.01 | 55.29 | 76.64 | 55.69 |
| ARMOR (2.5-32B ) | 2:4+3.44% | **78.18** | **78.77** | **76.56** | **39.51** | **60.15** | **79.32** | **59.78** |
| SparseGPT (2.5-72B) | 2:4 | 78.71 | 72.27 | 75.31 | 27.46 | 62.29 | 80.03 | 58.71 |
| Wanda (2.5-72B) | 2:4 | 79.61 | 75.66 | 75.96 | 23.88 | 62.37 | 80.19 | 59.43 |
| NoWag-P (2.5-72B) | 2:4 | 78.93 | 75.13 | 76.04 | 28.35 | 60.84 | 79.08 | 59.31 |
| ARMOR (2.5-72B ) | 2:4+2.4% | **82.40** | **82.11** | **79.42** | **40.40** | **63.40** | **80.90** | **62.64** |

Table 1: Results of Qwen-2.5 7B/14B/32B/72B. The additional $o\%$ for ARMOR pruned models represent the relative overhead of the block diagonal matricies. ARC-C is short for ARC-Challenge, Wino is short for WinoGrande, Hella is short for HellaSwag. (2.5-7B) denotes Qwen 2.5-7B, (2.5-14B) denotes Qwen 2.5-14B, etc.

| Method | Sparsity | Task Accuracy (%) (↑) | | | | | | |
|---|---|---|---|---|---|---|---|---|
| | | MMLU | GSM8K | BBH | GPQA | ARC-C | Wino | Hella |
| Dense (3-8B) | 0 | 76.82 | 84.91 | 77.42 | 42.86 | 63.65 | 76.87 | 58.92 |
| Dense (3-14B) | 0 | 80.47 | 84.0 | 78.50 | 39.29 | 66.64 | 79.08 | 61.81 |
| SparseGPT (3-8B) | 2:4 | 55.77 | 33.36 | 52.96 | 32.14 | 44.54 | 66.46 | 44.74 |
| Wanda (3-8B) | 2:4 | 55.75 | 27.45 | 46.32 | 29.46 | 41.55 | 62.75 | 42.92 |
| NoWag-P (3-8B) | 2:4 | 54.1 | 28.28 | 43.37 | 28.57 | 40.02 | 61.48 | 42.78 |
| Ours (3-8B) | 2:4+5.03% | **66.22** | **50.8** | **60.13** | **33.93** | **50.34** | **68.59** | **49.55** |
| SparseGPT (3-14B) | 2:4 | 64.73 | 48.22 | 61.5 | 27.9 | 53.58 | 71.27 | 50.22 |
| Wanda (3-14B) | 2:4 | 62.93 | 52.16 | 57.53 | 29.02 | 50.51 | 69.14 | 48.61 |
| NoWag-P (3-14B) | 2:4 | 61.69 | 46.63 | 56.11 | 27.46 | 48.89 | 68.27 | 48.42 |
| Ours (3-14B ) | 2:4+3.89% | **71.43** | **63.38** | **68.28** | **29.91** | **56.31** | **74.35** | **53.77** |

Table 2: Results of Qwen-3 8B/14B Base. Same setup as the Qwen 2.5 results, Once again (3-8B) denotes Qwen-3 8B etc

adopt the baselines' evaluation protocol: reporting Wikitext-2 perplexity at a sequence length of 2048 (in contrast to the native 4096/8192 used in our main results in Table 3). We exclude MaskLLM (Fang et al., 2024) as it operates in a retraining regime requiring orders of mangitude more data and compute than ARMOR. We also omit Targeted Low-Rank Refinement (Shen et al., 2025) due to the lack of publicly available code. As shown in Table 5, ARMOR significantly outperforms RotPruner across all models. Against the concurrent work DenoiseRotator, ARMOR consistently outperforms the Wanda-based variant and remains highly competitive with the SparseGPT variant, surpassing it on Llama-2 7B and 13B. Crucially, ARMOR achieves this parity without the rigid inference constraints of rotation-based methods. While DenoiseRotator enforces a fixed computational overhead

| Method | Sparsity | Wikitext 2 (↓) | | | | | C4 (↓) | | | | |
|---|---|---|---|---|---|---|---|---|---|---|---|
| | | 2-7B | 2-13B | 2-70B | 3-8B | 3-70B | 2-7B | 2-13B | 2-70B | 3-8B | 3-70B |
| Dense | 0% | 5.12 | 4.57 | 3.12 | 5.54 | 2.58 | 6.63 | 6.05 | 4.97 | 7.10 | 5.78 |
| SparseGPT | 2:4 | 10.16 | 8.39 | 5.39 | 14.18 | 8.65 | 11.98 | 10.22 | 7.20 | 13.88 | 9.27 |
| Wanda | 2:4 | 11.35 | 8.36 | 5.20 | 22.42 | 8.29 | 13.80 | 10.96 | 7.19 | 21.63 | 9.63 |
| NoWag-P | 2:4 | 11.14 | 8.28 | 5.17 | 24.0 | 7.52 | 13.91 | 11.05 | 7.23 | 23.5 | 9.18 |
| ARMOR | 2:4+o | **7.21** | **6.37** | **4.55** | **10.10** | **5.95** | **9.36** | **8.59** | **6.44** | **11.22** | **7.50** |

Table 3: Wikitext 2 and C4 perplexities on Llama-2 7B/13B/70B and Llama-3 8B/70B. Perplexity evaluations performed at 4096 and 8192 context length for Llama-2 and Llama-3 models respectively. $o$ denotes the relative overhead of the block diagonal matricies, which is $4.94\%$, $3.95\%$, $2.42\%$ for the 7B, 13B, and 70B parameter models respectively. Following established notation 2-7B denotes Llama-2 7B etc

| | Qwen 2.5 7B | | | Qwen 2.5 14B | | | Batched MatVec |
|---|---|---|---|---|---|---|---|
| | Tokens/s | Max VRAM | Model Size | Tokens/s | Max VRAM | Model Size | (ms) |
| Dense | 4461 | 32.84GB | 14.23GB | 2013 | 41.13GB | 27.65GB | 9.04 |
| 2:4 | 5430 (**1.217x**) | 27.52GB | 8.89GB | 2157 (**1.071x**) | 30.29GB | 16.81GB | 4.85 (**1.86x**) |
| ARMOR | 5090 (**1.141x**) | 28.11GB | 9.25GB | 2096 (**1.041x**) | 31.32GB | 17.85GB | 5.77 (**1.57x**) |

Table 4: Inference speed, Max VRAM, and Model Size for Dense, naive 2:4 pruning, and ARMOR pruned Qwen 2.5 7B/14B. Qwen2.5 7B and Qwen2.5 14B generation was performed at batch size 2048 and 512 respectively. Rightmost column lists the timings and speedup of batched Matrix Vector multiplication between a batch of 8192 input activations and a dense, 2:4 pruned, and ARMOR factorized matrix for a standard `gate_proj` layer of Qwen 2.5 14B.

via orthogonal matrices, ARMOR offers a tunable trade-off between accuracy and latency through the block size hyperparameter $d_{\text{block}}$.

## 4.4 INFERENCE EFFICIENCY

To quantify the practical efficiency of ARMOR, we implemented a PyTorch only proof of concept and benchmarked its generation speed, max VRAM, and model size for Qwen 2.5 7B and 14B (Table 4), alongside batched Matrix-Vector timing (batch size 8192) for a Qwen 2.5-14B layer. ARMOR explicitly trades a small fraction of theoretical throughput for improved model quality: for this layer, ARMOR's block diagonal wrappers introduce a 3.4% additional flop overhead, reducing the theoretical maximum speedup from $2.0\times$ for naive 2:4 to $\sim 1.87\times$. Our implementation achieves $\sim 84\%$ of this theoretical limit, compared to $\sim 93\%$ for highly optimized native 2:4 kernels. In other words, even using an unoptimized proof of concept, ARMOR retains the majority of the speed and memory benefits of 2:4 sparsity while delivering state-of-the-art recovery.

## 4.5 EXTENSIONS TO N:M AND UNSTRUCTURED

ARMOR extends naturally to general N:M patterns. Furthermore, while the sparse core update is not computationally tractable for general unstructured sparsity, ARMOR can be applied to unstructured sparsity as well by only performing the continuous update step. In table 6 we present experiments comparing ARMOR with NoWag-P for 50% unstructured, 4:8, 5:8, and 6:8 patterns Llama-2 7B/13B. ARMOR consistently and significantly outperforms NoWag-P across all sparsity patterns and types. These experiments were ran with significantly less iterations, 5000 for unstructured, 2000 for semi-structured, than the main 2:4 results, as a result they only serve as a *lower bound* on ARMOR's performance on sparsity structures beyond 2:4 sparsity.

## 4.6 ABLATIONS

To validate our proxy loss choice, we track its relative value against the model's relative C4 perplexity during 20,000 iterations of the ARMOR proxy loss optimization algorithm. The left plot

| Method | Wikitext 2 (↓) | | | |
|---|---|---|---|---|
| | 2-7B | 2-13B | 2-70B | 3-8B |
| Dense | 5.47 | 4.88 | 3.32 | 6.13 |
| RotPruner | 9.20 | - | - | 11.65 |
| Wanda + DenoiseRotator | 9.88 | 8.77 | 4.85 | 11.41 |
| SparseGPT + DenoiseRotator | 8.70 | 6.81 | **4.64** | **10.01** |
| ARMOR | **7.65** | **6.80** | 4.80 | 11.27 |

Table 5: Comparison of Wikitext 2 perplexity (context length = 2048) of 2:4 pruned Llama-2 7B/13B/70B and Llama-3 8B between ARMOR, RotPruner and DenoiseRotator methods. Same ARMOR models as Table 3 evaluated at shorter context length.

| Method | Sparsity | Wikitext 2 (↓) | | C4 (↓) | |
|---|---|---|---|---|---|
| | | 2-7B | 2-13B | 2-7B | 2-13B |
| Dense | 0% | 5.12 | 4.57 | 6.63 | 6.05 |
| NoWag-P | 50% | 6.37 | 5.49 | 8.27 | 7.35 |
| ARMOR | 50%+$o$ | **6.13** | **5.42** | **8.01** | **7.23** |
| NoWag-P | 4:8 | 8.04 | 6.47 | 10.17 | 8.67 |
| ARMOR | 4:8+$o$ | **6.74** | **5.96** | **8.78** | **7.98** |
| NoWag-P | 5:8 | 5.89 | 5.14 | 7.61 | 6.80 |
| ARMOR | 5:8+$o$ | **5.62** | **5.02** | **7.29** | **6.67** |
| NoWag-P | 6:8 | 5.35 | 4.76 | 6.92 | 6.27 |
| ARMOR | 6:8+$o$ | **5.26** | **4.72** | **6.80** | **6.23** |

Table 6: Comparison of ARMOR vs. NoWag-P across 50% and various N:M sparsity patterns on Llama-2 7B/13B. Relative overhead $o$ is the same as Table 3.

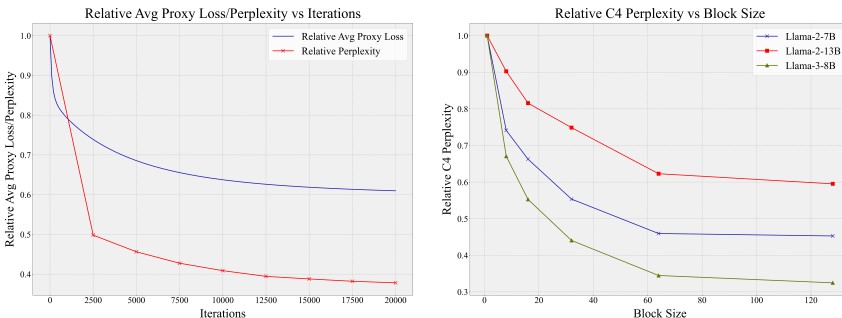

Figure 3: **Left:** Relative average Proxy Loss and C4 Perplexity of Llama-2 7B across 20,000 iterations of the ARMOR Proxy Loss optimization algorithm with block size 128. **Right:** Relative C4 Perplexity for Lama-2 7B/13B, and Llama-3 8B across block sizes of 1 (NoWag-P), 8, 16, 32, 64, and 128. ARMOR was ran for 5000 iterations for each block size. Relative perplexity is with respect to initial and optimal (dense) perplexities.

in Figure 3 shows a strong correlation between the two metrics; as the proxy loss decreases, so does perplexity, confirming its utility as a surrogate for overall model performance. Furthermore, we observe that the majority of the performance loss reduction was achieved within the first 2,500 iterations. We also performed an ablation study on block size to understand its impact on perplexity (Figure 3, right). This ablation reveals a clear trend across all models: increasing the block size improves performance by lowering perplexity in an exponential decaying manner. Additional ablations are detailed in Appendix E.

## 5 CONCLUSION

In this work, we introduce ARMOR, a novel one-shot algorithm that addresses the significant performance degradation of hardware-accelerated 2:4 pruning. Instead of simply removing weights, ARMOR reframes the problem by factorizing each weight matrix into a 2:4 sparse core and adaptive, low-overhead block diagonal wrappers that act as error correctors. This approach is theoretically guaranteed to converge to a solution with a proxy loss less than or equal to state-of-the-art methods and is empirically validated on Llama and Qwen family models, where it consistently and significantly outperforms existing 2:4 pruning techniques on both perplexity and downstream tasks. Crucially, ARMOR achieves these accuracy gains while retaining the majority of the inference speedups and memory reduction of native 2:4 sparsity. Our work demonstrates that rethinking weight representation is a powerful path toward establishing a more effective trade-off between the performance and efficiency of large language models.

## 6    REPRODUCIBILITY STATEMENT

Our code for ARMOR is available on Github at `https://github.com/LawrenceRLiu/ARMOR/tree/main`. To facilitate replication, we have also provided detailed descriptions of our architecture, algorithms, hyperparameter settings, and experimental setup in Appendix H.

## 7    ACKNOWLEDGMENTS

LY and LL are supported in part by NSF Grant 2221871. LY is also supported by an Amazon Faculty Award.

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

## A    NOTATION IN DEPTH

As established in section 3.1, for a block diagonal matrix $D \in \mathbb{R}^{d \times d}$ with block size $d_{\text{block}}$, we denote the individual blocks as:

$$
D = \begin{bmatrix}
D^{(1)} & 0 & \cdots & 0 \\
0 & D^{(2)} & \cdots & 0 \\
\vdots & \vdots & \ddots & \vdots \\
0 & 0 & \cdots & D^{\left(\frac{d}{d_{\text{block}}}\right)}
\end{bmatrix}
$$

where $D^{(i)} \in \mathbb{R}^{d_{\text{block}} \times d_{\text{block}}} \quad \forall \quad i \in [1, d/d_{\text{block}}]$. We choose a block diagonal wrappers over diagonal only because diagonal only offers no additional expressivity because it commutes with the element-wise pruning mask, mathematically:

$$
\text{diag}(a)(W' \odot M)\text{diag}(b) = W'' \odot M, \quad \forall \quad a \in \mathbb{R}^{d_{\text{in}}} \quad b \in \mathbb{R}^{d_{\text{out}}} \tag{5}
$$

where $W'' = \text{diag}(a)W'\text{diag}(b)$. In other words, Block-Diagonal matrices are a minimal, hardware acceleratable, structure that enable flexibility, which diagonal only matrices do not.

More generally for a matrix $C \in \mathbb{R}^{d_1 \times d_2}$ we denote the $d_{\text{block}} \times d_{\text{block}}$ matrix blocks as:

$$
C = \begin{bmatrix}
C^{(1,1)} & C^{(1,2)} & \cdots & C^{\left(1, \frac{d_2}{d_{\text{block}}}\right)} \\
\vdots & \vdots & \ddots & \vdots \\
C^{\left(\frac{d_1}{d_{\text{block}}}, 1\right)} & C^{\left(\frac{d_1}{d_{\text{block}}}, 2\right)} & \cdots & C^{\left(\frac{d_1}{d_{\text{block}}}, \frac{d_2}{d_{\text{block}}}\right)}
\end{bmatrix}
$$

## B    ARMOR OPTIMIZATION ALGORITHM IN DEPTH

In section B.1, we elaborate on the sparse core update step more formally. Additionally we provided pseudo code for both the sparse core update and the continuous parameter update steps in B.2.

### B.1    THE SPARSE CORE UPDATE STEP IN DEPTH

Decomposing equation 2 into independent subproblems at the block matrix level:

$$
\mathcal{L}_{W,X}(\theta) = \sum_{i=1} \sum_{j=1} \ell_X^{(i,j)}(\theta^{(i,j)}) = \sum_{i=1} \sum_{j=1} \left\| \bar{W}^{(i,j)} - A^{(i)} \left( W'^{(i,j)} \odot M^{(i,j)} \right) B^{(j)} \right\|_{F, \text{diag}(XX^T)^{(j)}}^2
$$
(6)

where $i \in [d_{\text{out}}/d_{\text{block}}]$ and $j \in [d_{\text{in}}/d_{\text{block}}]$. We optimize each block subproblem in parallel, for the remainder of this section we will consider block $(i,j)$. Optimizing these subproblems still requires a sweep over a exponential search space, so we adopt a greedy approach that optimizes a single sparse group for each subproblem.

**Optimizing the Sparse Group** After selecting group $i', k$, we freeze all values beyond $(W' \odot M)^{(i,j)}_{i',[k]}$, which we optimize with the 2:4 constraint. This is computationally feasible since we only have to sweep over $\binom{4}{2} = 6$ possible mask choices. Let us consider one such mask $m \in \{0, 1\}^4$ s.t. $\|m\|_0 = 2$, which has unmasked indices $i_1, i_2 \in [4]$ ie $m_{i_1} = m_{i_2} = 1$. Let us denote the unmasked elements for this mask as $w_m \mathbb{R}^2$, and $k' = 4(k-1)$. For this mask choice, the corresponding best-case proxy loss is given by:

$$
\ell_X^{(i,j)}(m) = \min_{w_m} \left\| \bar{W}^{(i,j)} - A^{(i)} W''^{(i,j)} B^{(j)} - A^{(i)}_{:,i'} w_m B^{(j)}_{\{k'+i_1, k'+i_2\}, :} \right\|_{F, \text{diag}(XX^T)^{(j)}}^2 \tag{7}
$$

where $W''^{(i,j)}$ represents the frozen remainder of the sparse core:

$$
W''^{(i,j)}_{n,p} = \begin{cases} 0 & \text{if } n = i', k'+1 \leq p \leq k'+4 \\ M^{(i,j)}_{n,p} W'^{(i,j)}_{n,p} & \text{otherwise} \end{cases} \quad \forall \quad n, p \in [d_{\text{block}}]
$$

Equation 7 can be arranged into a linear least squares problem with a closed form solution. For ease of notation, let $\Delta W = \bar{W}^{(i,j)} - A^{(i)} W''^{(i,j)} B^{(j)}$, $B' = B_{\{k'+i_1, k'+i_2\}, :}^{(j)}$, $a = A_{:,i'}^{(i)}$ and $D^{(j)} = \mathrm{diag}\left(XX^T\right)^{(j)}$. Solving the least squares problem results in an optimal $\ell_m^*$

$$\ell_m^* = \|\Delta W\|_{F, D^{(j)}}^2 - \frac{1}{\|a\|_2^2} \left(B' D^{(j)} \Delta W^T a\right)^T \left(B' D^{(j)} B'^T\right)^\dagger \left(B' D^{(j)} \Delta W^T a\right) \tag{8}$$

The first term is independent of the mask choice $m$ for the group, thus when sweeping over the possible masks, we only compute the second term for efficiency. The optimum is achieved with unmasked weights $w_m^*$:

$$w_m^* = \frac{1}{\|a\|_2^2} \left(B' D^{(j)} B'^T\right)^\dagger \left(B' D^{(j)} \Delta W^T a\right) \tag{9}$$

After sweeping over all 6 possible masks, we select the mask $m^*$, with nonzero values at $i_1^*, i_2*$, that achieves the minimum best-case proxy loss as given by equation 8. We substitute the corresponding optimal $w_{m^*}^*$ given by equation 9 into $W'$ to fully update sparse core, i.e. we set $\left(W_{\mathrm{new}}'^{(i,j)}\right)_{i',k'+i_1^*} = w_1$ and $\left(W_{\mathrm{new}}'^{(i,j)}\right)_{i',k'+i_2^*} = w_2$.

The 2:4 semi-structured sparsity allows for efficient calculation of equations 8 and 9. Since there are only 2 unmasked values for each group $\left(B' D^{(j)} B'^T\right)$ and $\left(B' D^{(j)} \Delta W^T a\right)$ are of shape $2 \times 2$ and $2 \times 1$. Thus the computational cost for solving for $\left(B' D^{(j)} B'^T\right)^\dagger \left(B' D^{(j)} \Delta W^T a\right)$ are negligible. Rather, calculating equations 8 and 9 are dominated by calculating $\left(B' D^{(j)} B'^T\right)$ and $\left(B' D^{(j)} \Delta W^T a\right)$, which scale with $d_{\mathrm{block}}^2$. We update all $d_{\mathrm{in}} d_{\mathrm{out}} / d_{\mathrm{block}}^2$ blocks at once, thus the overall computational complexity is $\mathcal{O}(d_{\mathrm{in}} d_{\mathrm{out}})$, scaling linearly with parameter size.

### B.2 ALGORITHM PSEUDOCODE

---

**Algorithm 2** Continuous Optimization via Sequential Gradient Descent

---

**Require:** Current parameters $A, B, W'$, mask $M$, normalized weight $\overline{W}$, calibration data $X$.
**Ensure:** Updated continuous parameters $A, B, W'$.
 1: **function** CONTINUOUSUPDATE($A, B, W', M, \overline{W}, X$)
 2:     $\hat{W} \leftarrow A(W' \odot M)B$
 3:     $L \leftarrow \|\overline{W} - \hat{W}\|_{F, diag(XX^T)}$            ▷ Compute proxy loss from Eq. 2
 4:                                      ▷ Update block diagonal matrix A
 5:     $\eta_A \leftarrow$ ComputeLearningRate($A, L$)        ▷ Based on local smoothness, Eq. 10
 6:     $A \leftarrow A - \eta_A \nabla_A L$
 7:                                      ▷ Update block diagonal matrix B
 8:     $\eta_B \leftarrow$ ComputeLearningRate($B, L$)                          ▷ Eq. 11
 9:     $B \leftarrow B - \eta_B \nabla_B L$
10:                                      ▷ Update underlying weight matrix W'
11:     $\eta_{W'} \leftarrow$ ComputeLearningRate($W', L$)                      ▷ Eq. 12
12:     $W' \leftarrow W' - \eta_{W'} \nabla_{W'} L$
13:     **return** $A, B, W'$
14: **end function**

---

---

**Algorithm 3** Greedy Sparse Core Update

---

**Require:** Current parameters $A, B, W'$, mask $M$, normalized weight $\overline{W}$, calibration data $X$.
**Ensure:** Updated sparse core parameters $W', M$.
 1: **function** SPARSECOREUPDATE($A, B, W', M, \overline{W}, X$)
 2:     **for** each block $(i, j)$ in parallel **do**
 3:                                                 ▷ Select a single 2:4 sparse group to update within the block
 4:         **for** each sparse group $(i', k)$ in block $(i, j)$ **do**
 5:             $p^{(i,j)}_{(i',k)} \propto \left\| \nabla_{(M \odot W')^{(i,j)}_{i',[k]}} \ell^{(i,j)} \right\|_1$             ▷ Calculate selection probability
 6:         **end for**
 7:         Select group $(i'^*, k^*)$ based on probabilities $p^{(i,j)}$.
 8:                                     ▷ Find the best mask and weights for the selected group
 9:         Let $\mathcal{M}$ be the set of all $\binom{4}{2} = 6$ possible masks for a group of 4.
10:         $l_{best} \leftarrow \infty$
11:         **for** each candidate mask $m \in \mathcal{M}$ **do**
12:             Calculate the resulting loss $l_m^*$ using Eq. 8
13:             **if** $l_m^* < l_{best}$ **then**
14:                 $l_{best} \leftarrow l_m^*$
15:                 $m^* \leftarrow m$
16:             **end if**
17:         **end for**
18:         Calculate optimal weights $w_m^*$ for the unmasked entries using Eq. 9
19:                               ▷ Update the mask M and weights W' for the chosen group
20:         Update mask for group $(i'^*, k^*)$ in $M^{(i,j)}$ with $m^*$.
21:         Update corresponding weights in $W'^{(i,j)}$ with $w_m^*$.
22:     **end for**
23:     **return** $W', M$
24: **end function**

---

## C   PROOFS

PROOF FOR PROPOSITION 1

**Proposition 1.** *The proxy loss is bounded below by* 0 *and is convex with respect to A, B, and $W'$ individually. A formal proof is provided in Appendix C*

*Proof.* The proxy loss is defined as:

$$\mathcal{L}_{W,X}(\theta) = ||\overline{W} - \hat{W}||^2_{F, \text{diag}(XX^T)} = \sum_{i,j} (\overline{W}_{ij} - \hat{W}_{ij})^2 ||X_j||^2_2$$

where $\hat{W} = A(W' \odot M)B$.

1. **Bounded Below by 0:** For each term in the summation, $(\overline{W}_{ij} - \hat{W}_{ij})^2 \geq 0$ as it is a squared real number. The weighting term $||X_j||^2_2$ is the squared Euclidean norm of a vector, which is also non-negative. A sum of non-negative terms is non-negative. Thus, $\mathcal{L}_{W,X}(\hat{W}) \geq 0$.

2. **Convexity with respect to A:** When B, $W'$, and M are held fixed, let $S = (W' \odot M)B$. The proxy loss function can be written as a function of A:

$$\mathcal{L}_{W,X}(A) = ||\overline{W} - AS||^2_{F,D}$$

where $D = \text{diag}(XX^T)$. This is a weighted least squares objective. The function $f(A) = \overline{W} - AS$ is an affine (linear) transformation of A. The function $g(Z) = ||Z||^2_{F,D}$ is a squared weighted Frobenius norm, which is a convex function. The composition of a convex function with an affine transformation is convex. Therefore, the proxy loss is convex with respect to A.

3. **Convexity with respect to B:** Similarly, when A, $W'$, and M are held fixed, let $S' = A(W' \odot M)$. The proxy loss function can be written as a function of B:

$$\mathcal{L}_{W,X}(B) = ||\overline{W} - S'B||^2_{F,D}$$

This is also a weighted least squares objective. Using the same reasoning as for A, the function is a composition of a convex function (squared norm) with an affine transformation of B, and thus is convex with respect to B.

4. **Convexity with respect to $W'$:** When A, B, and M are held fixed, the reconstructed weight matrix $\hat{W} = A(W' \odot M)B$ is a linear function of the elements of $W'$. Let $w' = \text{vec}(W')$, the vector of elements in $W'$. Then $\text{vec}(\hat{W})$ is a linear transformation of $w'$. The objective function $\mathcal{L}_{W,X}(\hat{W})$ is a quadratic function of the elements of $\hat{W}$, and therefore a quadratic function of the elements of $W'$. Specifically, it is a positive semidefinite quadratic form, which is convex. Therefore, the proxy loss is convex with respect to $W'$.

This completes the proof. □

PROOF FOR LEMMA C.1

**Lemma C.1.** *The continuous parameter update step results in a equal or lower proxy loss:*

$$\mathcal{L}_{W,X}\left(\hat{W}\left((A)_{t+1}, (B)_{t+1}, (W')_{t+1|\text{cont}}, (M)_t\right)\right) \leq \mathcal{L}_{W,X}\left(\hat{W}\left((A)_t, (B)_t, (W')_t, (M)_t\right)\right).$$

*Proof.* The paper states that the continuous optimization step updates the parameters A, B, and $W'$ using sequential gradient descent (Algorithm 2). The process is iterative:

1. Update A: $(A)_{t+1} = (A)_t - \eta_A \nabla_A \mathcal{L}_{W,X}((A)_t, (B)_t, (W')_t, (M)_t)$

2. Update B: $(B)_{t+1} = (B)_t - \eta_B \nabla_B \mathcal{L}_{W,X}((A)_{t+1}, (B)_t, (W')_t, (M)_t)$

3. Update $W'$: $(W')_{t+1|\text{cont}} = (W')_t - \eta_{W'} \nabla_{W'} \mathcal{L}_{W,X}((A)_{t+1}, (B)_{t+1}, (W')_t, (M)_t)$

From Proposition 1, the loss function $\mathcal{L}_{W,X}$ is convex with respect to each of A, B, and $W'$ individually. As Algorithm 2 states, the learning rate $\eta$ is determined via local $\beta$-smoothness, which are calculated in D.

For a convex and $\beta$-smooth function $f(x)$, the gradient descent update $x_{k+1} = x_k - \eta \nabla f(x_k)$ with a step size $0 < \eta \leq 1/\beta$ guarantees that $f(x_{k+1}) \leq f(x_k)$.

Applying this property to each step of the sequential update:

1. The update of A ensures that

$$\mathcal{L}_{W,X}((A)_{t+1}, (B)_t, (W')_t, (M)_t) \leq \mathcal{L}_{W,X}((A)_t, (B)_t, (W')_t, (M)_t)$$

2. The update of B, starting from the new A, ensures that

$$\mathcal{L}_{W,X}((A)_{t+1}, (B)_{t+1}, (W')_t, (M)_t) \leq \mathcal{L}_{W,X}((A)_{t+1}, (B)_t, (W')_t, (M)_t)$$

3. The update of $W'$, starting from the new A and B, ensures that

$$\mathcal{L}_{W,X}((A)_{t+1}, (B)_{t+1}, (W')_{t+1|\text{cont}}, (M)_t) \leq \mathcal{L}_{W,X}((A)_{t+1}, (B)_{t+1}, (W')_t, (M)_t)$$

Chaining these inequalities together, we get:

$$\mathcal{L}_{W,X}((A)_{t+1}, (B)_{t+1}, (W')_{t+1|\text{cont}}, (M)_t) \leq \mathcal{L}_{W,X}((A)_{t+1}, (B)_{t+1}, (W')_t, (M)_t)$$
$$\leq \mathcal{L}_{W,X}((A)_{t+1}, (B)_t, (W')_t, (M)_t) \leq \mathcal{L}_{W,X}((A)_t, (B)_t, (W')_t, (M)_t)$$

Thus, each continuous optimization step is guaranteed to not increase the proxy loss. □

PROOF FOR LEMMA C.2

**Lemma C.2.** *The sparse core update step results in a equal or lower proxy loss:*

$$\mathcal{L}_{W,X}\left(\hat{W}\left((A)_{t+1}, (B)_{t+1}, (W')_{t+1}, (M)_{t+1}\right)\right) \leq \mathcal{L}_{W,X}\left(\hat{W}\left((A)_{t+1}, (B)_{t+1}, (W')_{t+1|\text{cont}}, (M)_t\right)\right).$$

*Proof.* The discrete optimization step (Algorithm 3) seeks to improve the sparse core $W' \odot M$. The total proxy loss is decomposable into independent subproblems for each matrix block $(i, j)$, as shown in Equation 6:

$$\mathcal{L}_{W,X}(\theta) = \sum_{i=1}^{d_{\text{out}}/d_{\text{block}}} \sum_{j=1}^{d_{\text{in}}/d_{\text{block}}} ||\overline{W}^{(i,j)} - A^{(i)}((W' \odot M)^{(i,j)})B^{(j)}||^2_{F,\text{diag}(XX^T)^{(j)}}$$

The algorithm updates a single 2:4 sparse group $(i', k)$ within each block $(i, j)$. Let's focus on one such update. Let the loss before the update for this specific group be $l_{before}$. The current mask for this group is $m_{old}$, which is one of the $\binom{4}{2} = 6$ possible masks.

The algorithm proceeds as follows:

1. For each of the 6 possible masks $m$ in the set of valid masks $\mathcal{M}$, it calculates the optimal weights $w^*_m$ that minimize the loss for that group, assuming that mask $m$ is chosen (Equation 9). This results in the best possible loss $l^*_m$ for that mask choice (Equation 8).

2. It then selects the mask $m^*$ that yields the minimum loss among all 6 possibilities: $l_{best} = \min_{m \in \mathcal{M}} l^*_m$.

3. The algorithm updates the mask for group $(i', k)$ to $m^*$ and its corresponding weights in $W'$ to $w^*_{m^*}$.

The loss with the original mask $m_{old}$ and its weights before the update is $l_{before}$. The optimized loss for this original mask, $l^*_{m_{old}}$, must be less than or equal to $l_{before}$, since Equation 7 finds the optimal weights for any given mask.

$$l^*_{m_{old}} \leq l_{before}$$

By definition, the selected mask $m^*$ is the one that minimizes this optimal loss across all 6 choices. Therefore, its loss $l_{best}$ must be less than or equal to the optimal loss for the old mask $l^*_{m_{old}}$.

$$l_{best} \leq l^*_{m_{old}}$$

Combining these, we have $l_{best} \leq l_{before}$. The update for the selected group can only decrease or maintain the proxy loss. Since this holds for the update in each block, and the block losses are additive, the total proxy loss after the discrete optimization step is guaranteed to be less than or equal to the loss before the step. □

PROOF FOR THEOREM 3.1 (CONVERGENCE OF THE ARMOR OPTIMIZATION ALGORITHM)

**Theorem 3.1.** *(Convergence of the ARMOR optimization algorithm). The sequence $\{\mathcal{L}_{W,X}((\theta)_t)\}_{t \geq 0}$ converges and $\mathcal{L}_{W,X}((\theta)_t) \leq \mathcal{L}_{W,X}((\theta)_0) \quad \forall \quad t > 0$.*

*Proof.* Let $L_t = \mathcal{L}_{W,X}((\theta)_t)$ be the value of the proxy loss at the end of iteration $t$. Each iteration of the ARMOR optimization algorithm consists of a continuous optimization step followed by a discrete optimization step.

From Lemma C.1, the continuous step does not increase the loss. Let the loss after the continuous step at iteration $t$ be $L_{t+1|cont}$. We have:

$$L_{t+1|cont} \leq L_t$$

From Lemma C.2, the sparse core step, which follows the continuous step, also does not increase the loss. Let the loss after the discrete step be $L_{t+1}$. We have:

$$L_{t+1} \leq L_{t+1|cont}$$

Combining these two results gives:

$$L_{t+1} \le L_t$$

This shows that the sequence of proxy losses $\{L_t\}_{t \ge 0}$ is monotonically non-increasing. By induction, this implies that for any $t > 0$, $L_t \le L_0$.

Furthermore, from Proposition 1, the proxy loss is bounded below by 0. Therefore, $\{L_t\}_{t \ge 0}$ is a monotonically non-increasing sequence that is bounded below. By the Monotone Convergence Theorem, any such sequence must converge to a limit.

Therefore, the sequence $\{\mathcal{L}_{W,X}((\theta)_t)\}_{t \ge 0}$ converges, and its value is always less than or equal to its initial value. $\qquad\square$

## D  BETA SMOOTHNESS OF PROXY LOSS

In this section we present the proofs that the proxy loss is $\beta$-smooth with respect to each of the parameters $A$, $B$, $W'$. And using them, derive the learning rates for the sequential gradient descent based continous optimization step.

### D.1  PROXY LOSS $\beta$-SMOOTH WITH RESPECT TO $A$

First let us consider the $\beta$-smoothness of the proxy loss for block $(i,j)$ with respect to $A^{(i)}$. We have that:

$$\ell_X^{(i,j)}\left(\theta^{(i,j)}\right) = \left\| \bar{W}^{(i,j)} - A^{(i)}(M^{(i,j)} \odot W'^{(i,j)})B^{(j)} \right\|_{F,\text{diag}(XX^T)^{(j)}}^2$$

Where $\theta^{(i,j)} = (A^{(i)}, B^{(j)}, W'^{(i,j)}, M^{(i,j)})$. We have that:

$$\nabla_{A^{(i)}} \ell_X^{(i,j)}\left(\theta^{(i,j)}\right) = 2A^{(i)}\left(S^{(i,j)} D^{(j)} S^{(i,j)T}\right) - 2\bar{W}^{(i,j)} D^{(j)} S^{(i,j)T}$$

Where $S^{(i,j)} = (M^{(i,j)} \odot W'^{(i,j)})B^{(j)}$ and $D^{(j)} = \text{diag}\left(XX^T\right)^{(j)}$. Consdier two $A_{(1)}, A_{(2)} \in \mathbb{R}^{d_{\text{block}} \times d_{\text{block}}}$. Let us denote $\theta_{(1)}^{(i,j)} = (A_{(1)}^{(i)}, B^{(j)}, W'^{(i,j)}, M^{(i,j)})$ and $\theta_{(2)}^{(i,j)} = (A_{(2)}^{(i)}, B^{(j)}, W'^{(i,j)}, M^{(i,j)})$. Then we have that: Then we have that:

$$\left\| \nabla_{A^{(i)}} \ell_X^{(i,j)}(\theta_{(1)}^{(i,j)}) - \nabla_{A^{(i)}} \ell_X^{(i,j)}(\theta_{(2)}^{(i,j)}) \right\|_F = 2\left\| \left(A_{(1)}^{(i)} - A_{(2)}^{\prime(i)}\right)\left(S^{(i,j)} D^{(j)} S^{(i,j)T}\right) \right\|_F$$
$$\le 2\left\| A_{(1)}^{(i)} - A_{(2)}^{\prime(i)} \right\|_F \left\| S^{(i,j)} D^{(j)} S^{(i,j)T} \right\|_F$$

Where the inequality follows from the submultiplicativity of the Frobenius norm. Thus we have that the proxy loss $\ell_X^{(i,j)}(\theta^{(i,j)})$ is $\beta$-smooth with respect to $A^{(i)}$ with $\beta_A^{(i,j)} = 2\left\| S^{(i,j)} D^{(j)} S^{(i,j)T} \right\|_F$. We now consider the overall proxy loss, for some $A_{(1)}, A_{(2)} \in \mathbb{R}^{d_{\text{out}} \times d_{\text{in}}}$. Let $\theta_{(1)} = (A_{(1)}, B, W', M)$ and $\theta_{(2)} = (A_{(2)}, B, W', M)$. Then we have that:

$$\left\| \nabla_A \mathcal{L}_{W,X}(\theta_{(1)}) - \nabla_A \mathcal{L}_{W,X}(\theta_{(2)}) \right\|_F = \left\| \sum_{i=1}^{\frac{d_{\text{out}}}{d_{\text{block}}}} \sum_{j=1}^{\frac{d_{\text{in}}}{d_{\text{block}}}} \nabla_{A^{(i)}} \ell_X^{(i,j)}(\theta_{(1)}^{(i,j)}) - \nabla_{A^{(i)}} \ell_X^{(i,j)}(\theta_{(2)}^{(i,j)}) \right\|_F$$

$$\le \sum_{i=1}^{\frac{d_{\text{out}}}{d_{\text{block}}}} \sum_{j=1}^{\frac{d_{\text{in}}}{d_{\text{block}}}} \left\| \nabla_{A^{(i)}} \ell_X^{(i,j)}(\theta_{(1)}^{(i,j)}) - \nabla_{A^{(i)}} \ell_X^{(i,j)}(\theta_{(2)}^{(i,j)}) \right\|_F$$

$$\le 2\left\| A_{(1)} - A_{(2)} \right\|_F \sum_{i=1}^{\frac{d_{\text{out}}}{d_{\text{block}}}} \sum_{j=1}^{\frac{d_{\text{in}}}{d_{\text{block}}}} \left\| S^{(i,j)} D^{(j)} S^{(i,j)T} \right\|_F$$

Where the first inequality follows from the triangle inequality and the second follows from the previous result. Therefore the overall loss $\mathcal{L}_{W,X}(\theta)$ is $\beta$-smooth with respect to $A$ with $\beta_A = 2\sum_{i=1}^{\frac{d_{\text{out}}}{d_{\text{block}}}}\sum_{j=1}^{\frac{d_{\text{in}}}{d_{\text{block}}}}\left\|S^{(i,j)}D^{(j)}S^{(i,j)T}\right\|_F$. This leads us to a learning rate of:

$$\eta_A = \frac{1}{\beta_A} = \frac{1}{2\sum_{i=1}^{\frac{d_{\text{out}}}{d_{\text{block}}}}\sum_{j=1}^{\frac{d_{\text{in}}}{d_{\text{block}}}}\left\|S^{(i,j)}D^{(j)}S^{(i,j)T}\right\|_F} \tag{10}$$

## D.2 Proxy Loss $\beta$-smooth With Respect to $B$

Once again, we start by considering the $\beta$-smoothness of the proxy loss for block $(i,j)$ with respect to $B^{(j)}$. Reusing the notation we introduced in section, we have that:

$$\nabla_{B^{(j)}}\ell_X^{(i,j)}(\theta^{(i,j)}) = 2S'^{(i,j)T}S^{(i,j)}B^{(j)}D^{(j)} - 2S'^{(i,j)T}\bar{W}^{(i,j)}D^{(j)}$$

Where $S'^{(i,j)} = A^{(i)}(M^{(i,j)} \odot W'^{(i,j)})$ and $D^{(j)} = \text{diag}\left(XX^T\right)^{(j)}$. Consider two $B_{(1)}, B_{(2)} \in \mathbb{R}^{d_{\text{block}} \times d_{\text{block}}}$. Let us denote $\theta_{(1)}^{(i,j)} = (A^{(i)}, B_{(1)}^{(j)}, W'^{(i,j)}, M^{(i,j)})$ and $\theta_{(2)}^{(i,j)} = (A^{(i)}, B_{(2)}^{(j)}, W'^{(i,j)}, M^{(i,j)})$. Then we have that:

$$\left\|\nabla_{B^{(j)}}\ell_X^{(i,j)}(\theta_{(1)}^{(i,j)}) - \nabla_{B^{(j)}}\ell_X^{(i,j)}(\theta_{(2)}^{(i,j)})\right\|_F = 2\left\|S'^{(i,j)T}S^{(i,j)}\left(B_{(1)}^{(j)} - B_{(2)}^{(j)}\right)D^{(j)}\right\|_F$$
$$\leq 2\left\|S'^{(i,j)T}S^{(i,j)}\right\|_F\left\|B_{(1)}^{(j)} - B_{(2)}^{(j)}\right\|_F\left\|D^{(j)}\right\|_F$$

Where the inequality follows from the submultiplicativity of the Frobenius norm. Thus we have that the proxy loss $\ell_X^{(i,j)}(\theta^{(i,j)})$ is $\beta$-smooth with respect to $B^{(j)}$ with $\beta_B^{(i,j)} = 2\left\|S'^{(i,j)T}S'^{(i,j)}\right\|_F\left\|D^{(j)}\right\|_F$. Following the same procedure as the $A$ case, consider some $B_{(1)}, B_{(2)} \in \mathbb{R}^{d_{\text{in}} \times d_{\text{in}}}$. Let $\theta_{(1)} = (A, B_{(1)}, W', M)$ and $\theta_{(2)} = (A, B_{(2)}, W', M)$. Then we have that for the overall proxy loss:

$$\left\|\nabla_B\mathcal{L}_{W,X}(\theta_{(1)}) - \nabla_B\mathcal{L}_{W,X}(\theta_{(2)})\right\|_F \leq 2\left\|B_{(1)} - B_{(2)}\right\|_F\sum_{i=1}^{\frac{d_{\text{out}}}{d_{\text{block}}}}\sum_{j=1}^{\frac{d_{\text{in}}}{d_{\text{block}}}}\left\|S'^{(i,j)T}S^{(i,j)}\right\|_F\left\|D^{(j)}\right\|_F$$

Therefore the overall loss $\mathcal{L}_{W,X}(\theta)$ is $\beta$-smooth with respect to $B$ with $\beta_B = 2\sum_{i=1}^{\frac{d_{\text{out}}}{d_{\text{block}}}}\sum_{j=1}^{\frac{d_{\text{in}}}{d_{\text{block}}}}\left\|S'^{(i,j)T}S^{(i,j)}\right\|_F\left\|D^{(j)}\right\|_F$. This leads us to a learning rate of:

$$\eta_B = \frac{1}{\beta_B} = \frac{1}{2\sum_{i=1}^{\frac{d_{\text{out}}}{d_{\text{block}}}}\sum_{j=1}^{\frac{d_{\text{in}}}{d_{\text{block}}}}\left\|S'^{(i,j)T}S^{(i,j)}\right\|_F\left\|D^{(j)}\right\|_F} \tag{11}$$

## D.3 Proxy Loss $\beta$-smooth With Respect to $W'$

Finally, we consider the $\beta$-smoothness of the proxy loss with respect to $W'$. We have that:

$$\nabla_{W'}\mathcal{L}_{W,X}(\theta) = \left(2A^TA(W' \odot M)B\text{diag}\left(XX^T\right)B^T - 2A^T\bar{W}\text{diag}\left(XX^T\right)B^T\right) \odot M$$

Consider two $W'_{(1)}, W'_{(2)} \in \mathbb{R}^{d_{\text{out}} \times d_{\text{in}}}$. Let us denote $\theta_{(1)} = (A, B, W'_{(1)}, M)$ and $\theta_{(2)} = (A, B, W'_{(2)}, M)$. Then we have that:

$$\left\|\nabla_{W'}\mathcal{L}_{W,X}(\theta_{(1)}) - \nabla_{W'}\mathcal{L}_{W,X}(\theta_{(2)})\right\|_F = 2\left\|\left(A^TA\left((W'_{(1)} - W'_{(2)}) \odot M\right)B\text{diag}\left(XX^T\right)B^T\right) \odot M\right\|_F$$
$$\leq 2\left\|A^TA\left((W'_{(1)} - W'_{(2)}) \odot M\right)B\text{diag}\left(XX^T\right)B^T\right\|_F$$
$$\leq 2\left\|A^TA\right\|_F\left\|B\text{diag}\left(XX^T\right)B^T\right\|_F\left\|(W'_{(1)} - W'_{(2)}) \odot M\right\|_F$$
$$\leq 2\left\|A^TA\right\|_F\left\|B\text{diag}\left(XX^T\right)B^T\right\|_F\left\|W'_{(1)} - W'_{(2)}\right\|_F$$

Where the first and third inequalities follow from the property that $\|V \odot M\|_F \leq \|M\|_{\max} \|V\|_F$ for any matrices $V, M$ of the same shape, and the second follows from the submultiplicativity of the Frobenius norm. Thus we have that the proxy loss $\mathcal{L}_{W,X}(\theta)$ is $\beta$-smooth with respect to $W'$ with $\beta_{W'} = 2 \left\|A^T A\right\|_F \left\|B \mathrm{diag}\left(XX^T\right) B^T\right\|_F$. This leads us to a learning rate of:

$$\eta_{W'} = \frac{1}{\beta_{W'}} = \frac{1}{2 \left\|A^T A\right\|_F \left\|B \mathrm{diag}\left(XX^T\right) B^T\right\|_F} \tag{12}$$

# E    ABLATIONS

## E.1    SELECTION HEURISTIC

We investigated four choices for the sparse group selection heuristic used in the Discrete Optimization steps:

- **Uniform Random Selection (Random)**. We select a sparse group from the available $d_{\mathrm{block}}^2/4$ sparse groups within a sparse core block.
- **L1 Greedy Selection**, We directly select the sparse group based on which group has the maximum L1 gradient norm.
- **L2 Random Selection**, We draw a sparse group with probability that is based on the L2 norm of the gradient, instead of the L1 norm:

$$p_{(i',k)}^{(i,j)} \propto \left\|\nabla_{(M \odot W')^{(i,j)}_{i',[k]}} \ell_X^{(i,j)}\right\|_2 \quad \forall \quad i' \in [d_{\mathrm{block}}], \quad k \in [d_{\mathrm{block}}/4].$$

- **L1 Random Selection**, This is the selection heuristic discussed in the main text and that we use.

The results of this ablation is listed in table 7. L1 random and L2 random perform roughly equivalently.

| Method | Wikitext 2 ($\downarrow$) | | C4 ($\downarrow$) | |
|---|---|---|---|---|
| | 2-7B | 2-13B | 2-7B | 2-13B |
| Random | 8.17 | 7.05 | 10.49 | 9.48 |
| L1 Greedy | 8.46 | 7.28 | 10.87 | 9.72 |
| L2 Random | 7.95 | 7.00 | 10.27 | 9.38 |
| L1 Random | 7.99 | 6.99 | 10.30 | 9.39 |

Table 7: C4 and Wikitext 2 perplexities of ARMOR pruned Llama-2-7B/13B for different sparse group selection heuristics. Block size of 128 was used, and ARMOR optimization was ran for 2000 iterations to expedite runtime.

## E.2    CALIBRATION DATASET

| Dataset | Wikitext 2 ($\downarrow$) | C4 ($\downarrow$) |
|---|---|---|
| RedPajama-Data-1T | 7.59 | 9.89 |
| SlimPajama-627B | 7.68 | 9.88 |

Table 8: Perplexity of ARMOR pruned Llama-2-7B on Wikitext2 and C4 Datasets when using RedPajama-Data-1T vs SlimPajama-627B for our calibration dataset. ARMOR is minimally sensitive to the choice of dataset, so long as it is representative of the pre-training dataset. Block size of 128 was used, and ARMOR optimization was ran for 5000 iterations.

## E.3    CALIBRATION DATASET SAMPLE SIZE

By default, we use 128 samples from the SlimPajama-627B dataset, with sample length = 4096 for Llama-2 and 8192 for Llama-3 and Qwen models. This amounts to 524.29K and 1.049M tokens

respectively. In table 9, we additionally evaluated the performance of ARMOR pruned Llama-2 7B using 16, 32, and 64 samples from SlimPajama-627B. There is $< 1\%$ change in both Wikitext 2 and C4 perplexity across the entire range of sample sizes, indicating that ARMOR is highly data efficient.

| Num Samples | Tokens | Wikitext 2 ($\downarrow$) | C4 ($\downarrow$) |
|---|---|---|---|
| 16 | 65.54K | 7.61 | 9.86 |
| 32 | 131.07K | 7.66 | 9.86 |
| 64 | 262.14K | 7.63 | 9.86 |
| 128 (Default) | 524.29K | 7.66 | 9.93 |

Table 9: Wikitext 2 and C4 perplexities of ARMOR pruned Llama-2 7B using various numbers of calibration dataset samples (sample length = 4096). Block size of 128 was used, and 5000 iterations were used for each number of samples

## F    EXTENSION TO MIXTURE OF EXPERT MODELS

ARMOR supports MoE models out-of-the-box. In Table 10, we applied ARMOR to the Qwen3-30B-A3B MoE model and evaluated it on MMLU, comparing against NoWag-P. Due to time constraints, we used a highly restricted setup: 1,000 iterations (vs. 20,000 for the main results presented in Tables 1,2, and 3) and block size 32 (vs. 128). We compared this against the NoWag-P baseline on the same model. To account for the sparse routing of MoE architectures, we increased the calibration dataset size to 512 samples (vs. 128) for both ARMOR and the baseline to ensure sufficient data coverage for all experts. ARMOR outperformed the NoWag-P baseline by +5.83%. Furthermore, the degradation of ARMOR on this MoE model ( 8.7%) is remarkably consistent with the degradation we observe on dense models (e.g.,  8.6% for Qwen-2.5-7B in Table 1). This indicates that the ARMOR factorization and optimization strategy is robust across both Dense and Mixture-of-Experts architectures.

| Method | MMLU (%) ($\uparrow$) | Gap ($\downarrow$) |
|---|---|---|
| Dense | 81.16 | - |
| NoWag-P | 66.67 | 14.5% |
| ARMOR | **72.50** | **8.67%** |

Table 10: ARMOR and NoWag 2:4 pruned Qwen3-30B-A3B MoE model performance on MMLU. Block size 32 and 1000 iterations were used for ARMOR

## G    FEW SHOT TASK DESCRIPTIONS

ARMOR prund Qwen2.5 and Qwen 3 models were evaluated on an industry standard suite of 7 few shot task benchmarks. The results were reported in Tables 2 and 1. A detailed description of each task is included below. HuggingFace accelerate (Gugger et al., 2022) was used for parallel processing, and the EleutherAI LM evaluation harness was used (Gao et al., 2024). `max_model_len` = 4096 for all benchmarks for speedup.

1. **MMLU** (Hendrycks et al., 2021b;a) — The Massive Multitask Language Understanding (MMLU) benchmark is a 57-subject multiple-choice benchmark spanning STEM, humanities, social sciences, and professional domains that evaluates broad knowledge and basic reasoning. We report the 5-shot accuracy. (`acc,none`).

2. **GSM8K** (Cobbe et al., 2021) — Grade School Math 8K (GSM8K) is a dataset of 8.5K high quality linguistically diverse grade school math word problems. Problems require no concepts beyond the level of early Algebra, and solutions are provided in natural language. We report the 8-shot strict match: (`exact_match,strict-match`).

3. **ARC-c** (Clark et al., 2018) — The challenge subset of the AI2 Reasoning Challenge (ARC), which is composed of grade-school science questions authored for human tests. The challenge subset is composed of questions that baseline algorithms have failed on. We report the 25-shot accuracy. (`acc,none`).

4. **HellaSwag** (Zellers et al., 2019) — The HellaSwag challenge dataset has models choose the most commonsense continuation of everyday scenarios among four options. The benchmark is intentionally challenging for models. We report the 0-shot acurracy `acc,none`.

5. **BBH** (Suzgun et al., 2022) — BIG-Bench Hard (BBH) is a suite of 23 especially challenging tasks from the Beyond the Imitation Game benchmark (BIG-Bench). These tasks require multi-step reasoning and compositional generalization. We report the 3 shot match (`exact_match,get-answer`).

6. **WinoGrande** (Sakaguchi et al., 2021) — WinoGrande is a large dataset of 44k problems inspired by those of the Winograd Schema Challenge, systematically designed to minimize bias. The questions are of a 2-choice fill in the blank format that tests commonsense reasoning. We report the 5-shot accuracy (`acc,none`).

7. **GPQA (Main Set)** (Rein et al., 2024) — The Graduate-Level Google-Proof Q&A benchmark (GPQA) main set is a multiple-choice benchmark of 448 graduate-level biology, physics, and chemistry questions authored by experts. The main set excludes questions that are likely to be not objective and difficult. We report the 5-shot accuracy (`acc,none`).

## H    IMPLEMENTATION DETAILS

We used a block size of 128 for all models. For optimization, we used a learning rate of $10^{-4}$ for ADAM, and ran for 20,000 iterations. We used 128 samples of the SlimPajama-627B dataset (Soboleva et al., 2023) as our calibration dataset. Each sample had a context length of 4096 for Llama-2 family models, and 8192 for Llama-3, Qwen 2.5, and Qwen 3 family models. Inference benchmarks were performed on a 48GB 4090.

## I    LLM USAGE

We used LLMs to polish and refine our language. Some icons, such as the robot in Figure 1 and the fire and snowflake icon in Figure 2 were generated with generative models. Furthermore, we used LLMs to conduct some literature review to supplement and double check our manual literature review.

