# OpenReview forum: "ARMOR: High-Performance Semi-Structured Pruning via Adaptive Matrix Factorization"
_ICLR.cc/2026/Conference — ICLR 2026 Poster_

### Official Review · Reviewer_G27K · 2025-10-21

**Soundness:** 3
**Presentation:** 3
**Contribution:** 3
**Rating:** 6
**Confidence:** 4

**Summary:**

In this paper, the authors address the trade-off between hardware acceleration and performance degradation in semi-structured pruning. They propose an adaptive matrix factorization framework that decomposes the original weight matrix W into three components (A, W’xM, B). Then they employ a block coordinate descent algorithm to iteratively optimize these parameters. Experimental results on LLaMA and Qwen demonstrate the effectiveness of the proposed approach compared to existing methods. The inference analysis further indicates that the method introduces a slight computational overhead relative to standard 2:4 pruning. Moreover, it provides a theoretical proof to guarantee the algorithm’s convergence.

Overall, the paper presents a decomposable and learnable 2:4 pruning scheme for LLMs, offering a meaningful alternative for LLM compression research. Therefore, I lean to marginally accept this paper, and I would be willing to raise my score if the authors address my main concerns during the rebuttal phase.

**Strengths:**

Originality:
This paper proposes a decomposable and learnable 2:4 pruning scheme for large language models (LLMs), which is both novel and valuable for LLM compression research. It provides a promising approach to enhance the performance of 2:4 pruning while preserving practical inference efficiency.

Quality:
The paper is well-written and well-organized, presenting its ideas in a coherent and logical manner.

Clarity:
The content is clearly presented, and the mathematical formulations are easy to read and follow.

Significance:
The study introduces an alternative approach to improving the performance of 2:4 pruning for LLMs, making it a meaningful contribution toward efficient and practical model deployment.

**Weaknesses:**

Main concerns:
1. The paper’s main weakness lies in limited novelty relative to existing learnable semi-structured pruning frameworks. The proposed method introduces learnable transformations (A, W' B) to improve 2:4 LLM pruning, but this concept is similar to prior learnable pruning works such as MaskLLM [1], RotPruner [2], and DenoiseRotator [3], which also learn rotation or masking matrices to enhance pruning robustness and recovery. To convincingly demonstrate novelty, the authors should explicitly position their method against these approaches—highlighting what differentiates their transformation learning (e.g., new optimization formulation, stability, or efficiency aspects).

2. The main tables compare against SparseGPT, Wanda, and NoWag, which are not the current state-of-the-art methods for semi-structured pruning. Without comparisons to MaskLLM, RotPruner, and DenoiseRotator, it is difficult to assess the real performance improvement or contribution of the proposed method.

3. It lacks insufficient explanation of why a block-diagonal structure is preferred over a simple diagonal form for A and B matrix.

4. the inference acceleration results appear underwhelming. For 2:4 sparsity, the theoretical speedup ratio should ideally approach 2×, but the reported gain is marginal. This suggests the need for more detailed discussion or optimization strategies to bridge the gap between theoretical sparsity and actual hardware acceleration.

Minor comments:
1. Line 78-79 syntax error, “Thus it is difficult to translate the theoretical model size reductions into practical inference speedups is difficult”


[1] MaskLLM: Learnable Semi-Structured Sparsity for Large Language Models. NeurIPS 2024.
[2] RotPruner: Large Language Model Pruning in Rotated Space.
[3] DenoiseRotator: Enhance Pruning Robustness for LLMs via Importance Concentration. NeurIPS 2025.

**Questions:**

1. Comparison with SOTA methods:
Can the authors include results comparing their approach with recent learnable semi-structured pruning methods such as MaskLLM [1], RotPruner [2], and DenoiseRotator [3]? This would help clarify whether the proposed learnable transformation provides measurable benefits beyond these baselines.

2. Generality across N:M patterns:
Does the proposed method extend naturally to other N:M pruning settings (e.g., 4:8, 5:8, or 6:8)? If so, how should hyperparameters like block size and iteration count be adjusted?

3. Structure of transformation matrices:
What is the rationale behind adopting a block-diagonal structure for matrices A and B? Would using a pure diagonal structure simplify computation while maintaining comparable performance?

4. Inference efficiency:
The observed inference speedup for 2:4 sparsity is much smaller than the theoretical 2× expectation. Are there software or hardware factors limiting this gain? Could kernel fusion or custom sparse-GEMM implementations improve it?

Minor issue:
Please correct the syntactic error at lines 78–79 (“...speedups is difficult”).

---

> ### Author Response · Authors · 2025-11-21
>
> We thank Reviewer G27K for their evaluation and for acknowledging the originality of our decomposable, learnable pruning scheme as a meaningful alternative for LLM compression. In addition we thank the reviewer for identifying the wording problem in lines 78-79, we have now corrected that in the updated manuscript (highlighted in red). We address the specific weaknesses and questions below:
>
> **W1, W2, and Q1: Additional Baselines:**
> We thank the reviewer for these references. To demonstrate ARMOR's novelty, we explicitly position it against these works based on computational regime and architectural scope:
>
> **Regime Difference (vs. MaskLLM):** ARMOR is a strict One-Shot method (no retraining), whereas MaskLLM operates in a Retraining regime, by parameterizing the sparse masks as gumble softmaxes and training them. Comparing ARMOR with MaskLLM is an apples-to-oranges comparison. **ARMOR uses 4000x less data** (128 samples vs 512k samples) and **80x less compute** (16 A6000 GPU hours vs 1280 A100 hours) than MaskLLM
>
> **Structural Difference (vs. Rotation Methods):** RotPruner and DenoiseRotator enhance sparsity by rotating the weight basis using orthogonal matrices. In contrast, ARMOR factorizes the weights using Block-Diagonal wrappers. This block-diagonal structure allows for a directly tunable overhead (via block size) vs the fixed overhead of Rotation methods. Furthermore, unlike RotPruner/DenoiseRotator, which rely on transformer-specific rotation strategies, ARMOR factorizes linear layers agnostically, making it compatible with any architecture using linear projections. Below we compare against RotPrune and DenoiseRotator individually:
>
> **RotPrune:** Below we have included a table comparing the Wikitext2 perplexities (seqlen=2048) of ARMOR and RotPrune on Llama-2 7B and Llama 3 8B. ARMOR significantly outperforms RotPruner on both. A full comparison is not possible because there is no public code for RotPrune.
>
> |  | Llama 2 7B | Llama 3-8B |
> | :---- | :---- | :---- |
> | **RotPruner:** | 9.20 | 11.65 |
> | **ARMOR** | **7.645** | **11.269** |
>
> **DenoiseRotator:** Below we have included a table comparing the Wikitext2 perplexities (seqlen=2048) of ARMOR and DenoisePruner pruned Llama-2 7B/13B/70B and Llama-3 8B on Wikitext 2\. ARMOR outperforms Wanda+DenoisePruner across all models and is competitive with SparseGPT+DenoisePruner. Additionally we would like to note that DenoiseRotator was accepted into Neurips, and as such would be classified by the ICLR guidelines as a contemporaneous work.
>
> |  | Llama 2 7B | Llama 2 13B | Llama 2 70B | Llama 3-8B |
> | :---- | :---- | :---- | :---- | :---- |
> | **Wanda \+ DenoiseRotator** | 9.876 | 8.772 | 4.845 | 11.411 |
> | **SparseGPT \+ DenoiseRotator** | 8.696 | 6.811 | **4.643** | **10.008** |
> | **ARMOR** | **7.645** | **6.80** | 4.80 | 11.269 |
>
> **W3 and Q3, Block Diagonal vs Diagonal Only:**
>
> We choose a block diagonal wrappers over diagonal only because diagonal only offers **no additional expressivity** because it commutes with the element-wise pruning mask, mathematically:
> $$\\text{diag}(a)(W'\\odot M)\\text{diag}(b)=W''\\odot M,$$
> where $W''=\\text{diag}(a)W'\\text{diag}(b)$. In other words, Block-Diagonal matrices are a minimal, hardware acceleratable, structure that enable flexibility, which diagonal only matrices do not.
>
> **Q2 Generality across N:M patterns:**
>
> ARMOR works out of the box for general N:M patterns. Below we have added an ablation comparing ARMOR with NoWag-P for 4:8, 5:8 and 6:8 patterns for Llama-2 7B/13B, comparing the Wikitext2 and C4 perplexities at native sequence length (seqlen=4096).
>
> |  |  | Wikitext2 PPL ($\downarrow$) | Wikitext2 PPL ($\downarrow$) | C4 PPL ($\downarrow$)| C4 PPL ($\downarrow$) |
> | :---- | :---- | ----- | :---- | ----- | :---- |
> |  | Sparsity | 2-7B | 2-13B | 2-7B | 2-13B |
> | NoWag-P | 4:8 | 8.04 | 6.47 | 10.17 | 8.67 |
> | ARMOR | 4:8 | **6.74** | **5.96** | **8.78** | **7.98** |
> | NoWag-P | 5:8 | 5.89 | 5.14 | 7.61 | 6.8 |
> | ARMOR | 5:8 | **5.62** | **5.02** | **7.29** | **6.67** |
> | NoWag-P | 6:8 | 5.35 | 4.76 | 6.92 | 6.27 |
> | ARMOR | 6:8 | **5.26** | **4.72** | **6.80** | **6.23** |
>
> For this experiment, ARMOR was run at block size 128 and for 2000 iterations, i.e. 1/10th of the iterations used for the main results (tables 1,2,3). Because both hyperparameters are directly tied to model performance (see fig 3), the same general rules apply. Increasing iterations results in a better performing model, at the cost of a longer compression runtime. Increasing the block size also results in a better performing pruned model, at the cost of more memory and inference overhead.

---

> > ### Author Response · Authors · 2025-11-21
> >
> > **W4 and Q4: Inference Speedup.**
> >
> > We appreciate the opportunity to clarify the inference efficiency. The reviewer notes that the speedup (1.57x) does not approach the theoretical 2x of standard pruning. This is expected behavior: **ARMOR effectively trades a mathematically bounded amount of theoretical speedup for significantly higher accuracy.**
> >
> > Unlike standard pruning (which reduces FLOPs by 50%), ARMOR introduces two block-diagonal wrapper multiplications. For a Qwen-2.5-14B gate projection layer (5120 by 13824\) with block size 128, the wrappers add a 3.4% flop overhead. This means that the theoretical maximum speedup for ARMOR is \~1.87x, compared to 2.0x for plain 2:4. Our measured speedup of **1.57x** achieves **84% of this theoretical maximum** (1.57/1.87). By comparison, the highly optimized plain 2:4 kernel (1.86x) achieves ~93% of its theoretical limit.
> >
> > The small efficiency gap (93% vs 84%) arises because our current implementation is Pytorch only, and uses three separate kernels (Wrapper to Sparse Core to Wrapper), incurring overhead from reading/writing intermediate activations to GPU memory. In other words, the gap the reviewer observes is not an implementation failure, but a precise reflection of the algorithmic trade-off (wrappers for accuracy) and the lack of operator fusion. We believe retaining **84% of the maximum possible efficiency while delivering state-of-the-art recovery is a highly practical contribution**.

---

> > ### Comment · Reviewer_G27K · 2025-11-24
> > **Official Comment by Reviewer G27K**
> >
> > Thank you for the authors’ detailed analysis and additional experiments. If the content presented in the rebuttal is included into the revised manuscript, I would be inclined to raise my final score.

---

> > > ### Author Response · Authors · 2025-11-24
> > >
> > > We thank reviewer G27K for their engagement and willingness to reevaluate our work. We have uploaded a revised manuscript that incorporates the additional experiments and analyses discussed in our rebuttal. We have highlighted all new additions in red.
> > >
> > > Below is a guide to where these changes can be found in the revision:
> > >
> > > 1. **Comparison with Learnable Baselines (MaskLLM, RotPruner, DenoiseRotator):** We have added a new section, **Appendix G**, which explicitly positions ARMOR against these methods. This section details the regime differences (one-shot vs. retraining) and structural differences, and includes **Tables 10 and 11** comparing ARMOR against RotPruner and DenoiseRotator on Llama-2 and Llama-3.
> > > 2. **Generality across N:M Patterns:** We have added **Section 4.4 (Extensions to N:M and Unstructured)** and **Table 5** to the main body. This demonstrates ARMOR’s effectiveness on 4:8, 5:8, and 6:8 sparsity patterns, as well as unstructured pruning.
> > > 3. **Inference Efficiency Analysis:** We have rewritten **Section 4.3 (Inference Efficiency)** to explicitly discuss the trade-off between theoretical throughput and accuracy. We clarify that ARMOR achieves \~84% of the theoretical maximum speedup (accounting for wrapper overhead).
> > > 4. **Block-Diagonal Rationale:** We have clarified the necessity of block-diagonal structures over diagonal-only wrappers in **Section 3.1** and **Appendix A**, noting that diagonal wrappers provide no additional expressivity as they commute with the element-wise mask.
> > > 5. **Minor Corrections:** The syntax error on Page 2 has been corrected.
> > >
> > > We believe these additions significantly strengthen the paper and address your concerns regarding novelty and efficiency. We hope this warrants an improved score.

---

> ### Comment · Reviewer_G27K · 2025-11-26
> **Official Comment by Reviewer G27K**
>
> I understand that there are page limits for the main text. However, I believe the comparison with learnable baselines should be included in Section 1.1 Related Work, while the comparison of ARMOR with RotPruner and DenoiseRotator (Tables 10 and 11) would be more appropriate in Section 4 Results.

---

> > ### Author Response · Authors · 2025-11-26
> >
> > We thank Reviewer G27K for their continued engagement. We agree that placing these comparisons in the main text significantly strengthens the paper's positioning. As such, we have updated the manuscript to reflect your suggestions:
> >
> > 1. **Section 1.1 (Related Work):** We have revised this section to explicitly categorize and contrast ARMOR against learnable masking (MaskLLM, LoSparse) and rotation-based methods (RotPruner, DenoiseRotator), highlighting the critical differences in computational regimes.
> >
> > 2. **Section 4.3 (Comparisons with Learnable Baselines):** We have moved the detailed empirical comparisons from the Appendix to this new main body subsection. This includes Table 5, which quantifies ARMOR's performance against RotPruner and DenoiseRotator, along with a discussion on why retraining-based methods (MaskLLM) are excluded from this specific benchmark.
> >
> > We believe this new structure provides the clarity and rigor you requested.

---

> > > ### Comment · Reviewer_G27K · 2025-11-27
> > > **Official Comment by Reviewer G27K**
> > >
> > > Thanks for the authors’ revision. I would like to raise my final score to 8.

---

> > > > ### Author Response · Authors · 2025-11-27
> > > >
> > > > We thank Reviewer G27K for their continued engagement and for raising their score. We are glad that our revisions and additional baselines satisfactorily addressed your concerns.

---

### Official Review · Reviewer_Ze6L · 2025-10-29

**Soundness:** 3
**Presentation:** 3
**Contribution:** 3
**Rating:** 6
**Confidence:** 3

**Summary:**

The paper proposes ARMOR, a post-training semi-structured pruning method for LLMs that factorizes each weight matrix into a 2:4 sparse core surrounded by block-diagonal transformation matrices (A and B) to mitigate accuracy loss.

**Strengths:**

1. The factorization approach is a novel way to add flexibility to rigid 2:4 pruning, potentially bridging the gap between unstructured and semi-structured methods.
2. The theoretical analysis in Appendix C is sound.

**Weaknesses:**

1. The proposed ARMOR method does not support unstructured pruning. It is specifically designed for semi-structured pruning (with a focus on 2:4 sparsity patterns), as the core optimization algorithm relies on the constrained group structure of semi-structured masks to remain computationally tractable. However, baseline methods such as Wanda support unstructured pruning.
2. The related work is sufficient. There are other refinement methods that also mitigate the gap between a dense model and a pruned model. For example, low-rank refinement [1, 2] and adaptive layer-wise sparsity control [3].
3. Minor concern: evaluations are restricted to dense models.

[1] Y. Li et al., “LoSparse: Structured Compression of Large Language Models based on Low-Rank and Sparse Approximation,” 2023.
[2] L. Shen, A. Tang, Y. Luo, T. Sun, H. Hu, and X. Cao, “Targeted Low-rank Refinement: Enhancing Sparse Language Model with Precision,”  2025.
[3] L. Yin et al., “Outlier Weighed Layerwise Sparsity (OWL): A Missing Secret Sauce for Pruning LLMs to High Sparsity,” 2024

**Questions:**

1. Does the proposed method support other models, such as MoE models? Does ARMOR degrade on them?

---

> ### Author Response · Authors · 2025-11-21
>
> We thank Reviewer Ze6L for their thoughtful review and for identifying our factorization approach as a novel means to bridge the gap between rigid N:M constraints and flexible representational capacity. Below we have addressed each of the reviewer’s weakness and questions in depth:
>
> **W1: Restriction To N:M Semistructured Sparsity.**
> The reviewer is correct that currently the sparse core optimization step leverages the N:M semi-structured constraint to remain tractable. However, ARMOR can be applied to unstructured sparsity by simply only performing the continuous optimization step. As a proof of concept we pruned Llama-2 7B/13B to with a 50% sparse core using ARMOR (block size 128, 5000 iterations). Below we have included a table comparing 50% NoWag-P and 50% ARMOR on Wikitext 2 and C4 perplexity (lower is better). **ARMOR demonstrates a significant improvement over NoWag-P for unstructured pruning as well.**
>
> |  | Wikitext 2 ($\\downarrow$) | Wikitext 2 ($\\downarrow$) | C4 ($\\downarrow$) | C4 ($\\downarrow$) |
> | :---- | ----- | :---- | ----- | :---- |
> |  | 2-7B | 2-13B | 2-7B | 2-13B |
> | NoWag-P 50% | 6.37 | 5.49 | 8.27 | 7.35  |
> | ARMOR 50% | **6.13** | **5.42** | **8.01** | **7.23** |
>
> These experiments are simply proofs of concept. Only 5000 iterations were run, rather than allowing the algorithm to fully converge. Likewise, sparse core update alternatives were not explored. Thus these results server only as a *lower bound* on ARMOR's performance at unstructured pruning.
>
> **W2: Related Work:**
> We assume the reviewer meant that our related works were *insufficient* rather than *sufficient*. We thank the reviewer for noting these works which we had overlooked, we have incorporated these into the related works section (new additions are highlighted in red). However we have also made specific comparisons against the related works noted.
>
> **LoSparse:** LoSparse is an iterative method requiring a modified training workflow where weights are gradually sparsified. In comparison, ARMOR is a strict one-shot compression algorithm, where a model is compressed in a single pass without any retraining.  In particular, our algorithm optimizes the wrapper-sparse-core matrix factorization for each linear projection independently. As a result, the compute requirements to compress a model with ARMOR is very small, we only need to be able to fit a single linear projection onto the device.
>
> **Targeted Low-rank Refinement:** We agree that this a highly relevant work, particularly in its exploration of enhancing sparse models with low-rank structures. We intended to conduct a detailed benchmark against this method; however, **there is currently no official code implementation publicly available.** Furthermore, the paper does not report critical reproducibility details such as the exact perplexity sequence lengths, prompt templates, or LM harness version used. As a result, conducting a fair and accurate empirical comparison within the rebuttal period was not feasible.
>
> **Outlier Weighted Layerwise Sparsity (OWL):** OWL specifically relies on unstructured sparsity to allocate different levels of sparsity to different layers. Due to the resulting unstructured memory access pattern, it is difficult to parallelize unstructured sparsity for GPU based acceleration. As a result the speedup experiments were CPU based acceleration evaluations through DeepSparse. OWL cannot be applied to N:M sparsity, which can be accelerated on GPUs, since N:M sparsity automatically enforces the same level of sparsity for each layer, In contrast ARMOR is specifically designed to leverage 2:4 sparsity to achieve a real world speedup on Nvidia GPUs. Future work could investigate allocating different block sizes (and thus overheads) to different layers based on some OWL “like” heuristic.

---

> ### Author Response · Authors · 2025-11-21
>
> **W3 (Minor) and Q1: MoE Models.**
>
> ARMOR supports MoE models out-of-the-box. To demonstrate this, we applied ARMOR to the Qwen3-30B-A3B MoE model and evaluated it on MMLU. Due to time constraints, we used a highly restricted setup: 1,000 iterations (vs. 20,000 in the paper's main results) and block size 32 (vs. 128). We compared this against the NoWag-P baseline on the same model. To account for the sparse routing of MoE architectures, we increased the calibration dataset size to 512 samples (vs. 128\) for both ARMOR and the baseline to ensure sufficient data coverage for all experts.
>
> |  | MMLU (%) ($\uparrow$) | Gap |
> | :---- | :---- | :---- |
> | Dense | 81.16 | \- |
> | NoWag-P | 66.67 | 14.49% |
> | ARMOR | **72.50** | **8.67%** |
>
> Even with significantly reduced optimization steps (1/20th of standard) and a smaller block size, ARMOR outperformed the NoWag-P baseline by \+5.82%. Furthermore, the degradation of ARMOR on this MoE model (\~8.7%) is remarkably consistent with the degradation we observe on dense models (e.g., \~8.6% for Qwen-2.5-7B in Table 1). This indicates that ARMOR's factorization and optimization strategy is robust across both Dense and Mixture-of-Experts architectures.

---

> > ### Author Response · Authors · 2025-11-24
> >
> > We thank reviewer Ze6L for their constructive feedback. We have uploaded a revised manuscript that incorporates the unstructured pruning and MoE experiments and the expanded literature review. These additions directly reflect the results discussed in our rebuttal.
> >
> > We have highlighted all new additions in the manuscript in red. Below is a guide to where these changes can be found in the revision:
> >
> > 1. **Extension to Unstructured Pruning (Addressing W1):** We have added **Section 4.4 (Extensions to N:M and Unstructured)** and **Table 5**, which demonstrate ARMOR's effectiveness on 50% unstructured sparsity as well as 4:8, 5:8, and 6:8 patterns.
> > 2. **Related Work (Addressing W2):** We have expanded **Section 1.1 (Related Work)** to explicitly discuss and contrast ARMOR against *LoSparse*, *Targeted Low-Rank Refinement*, and *OWL*. We highlight that ARMOR is a strict one-shot method (unlike LoSparse) and compatible with hardware-accelerated N:M patterns (unlike OWL).
> > 3. **Support for MoE Models (Addressing Q1 & W3):** We have added **Appendix F (Extension to Mixture of Expert Models)**, detailing experiments on the Qwen3-30B-A3B MoE model that show ARMOR functions out-of-the-box for sparse architectures.
> >
> > We would appreciate your feedback on whether these new results and revisions adequately resolve all of your concerns.

---

> > > ### Author Response · Authors · 2025-11-26
> > >
> > > We thank Reviewer Ze6L again for their valuable feedback. We wanted to provide a quick update regarding our latest manuscript revision, which incorporates feedback from all reviewers.
> > >
> > > * **Expanded Related Work (Section 1.1)**: We have reworked this section to explicitly discuss the works you mentioned (LoSparse, Targeted Low-Rank Refinement, OWL), as well as the learnable pruning baselines suggested by Reviewer G27K (MaskLLM, RotPruner,  DenoiseRotator).
> > > * **Clarification on Baselines:** We added a note in Section 4.3 clarifying that Targeted Low-Rank Refinement could not be empirically benchmarked due to the lack of public code/reproducibility details.
> > > * **Consolidated Experiments:** As a reminder, the unstructured pruning results (addressing W1) and MoE Qwen-3 results (addressing W3 (Minor) and Q1) are fully detailed in Section 4.5 and Appendix F, respectively.
> > >
> > > We believe these revisions, alongside the MoE and Unstructured experiments previously discussed, comprehensively address your concerns regarding model versatility and related work. We would appreciate your feedback on whether these updates resolve your reservations.

---

> > > > ### Comment · Reviewer_Ze6L · 2025-11-27
> > > >
> > > > Thank you for the detailed response and updated results! My concerns have been addressed, and I will raise my rating.

---

### Official Review · Reviewer_4fMQ · 2025-11-01

**Soundness:** 4
**Presentation:** 4
**Contribution:** 4
**Rating:** 8
**Confidence:** 4

**Summary:**

The paper proposes ARMOR, a one-shot 2:4 semi-structured pruning method for large language models. Instead of directly pruning weights, each matrix is factorized into a 2:4 sparse core surrounded by two block-diagonal “wrapper” matrices that act as lightweight error correctors. The method is optimized via a block coordinate descent alternating between gradient updates for the wrappers and greedy least-squares updates for the sparse core. ARMOR achieves large accuracy gains over prior 2:4 pruning methods (Wanda, SparseGPT, NoWag-P) while keeping almost identical inference speedups and memory savings. This is an excellent paper that combines a simple yet powerful idea with strong empirical validation and theoretical rigor. It clearly advances the state of the art in semi-structured pruning, a practically important domain for LLM deployment.

**Strengths:**

1. Recasts semi-structured pruning as a matrix-factorization problem instead of a masking problem, providing a new theoretical and practical perspective. The idea of block-diagonal wrappers acting as error-correcting transformations is elegant and hardware-friendly.
2. Consistent, substantial improvements over multiple strong baselines (Wanda, SparseGPT, NoWag-P). Performance nearly matches dense models on some reasoning benchmarks.
3. Provides a convergence guarantee showing that ARMOR’s proxy loss never exceeds that of the initialization (NoWag-P). Proxy loss design is interpretable and data-aware (weighted Frobenius norm).
4. Includes both perplexity and task-based evaluation, block-size ablation, and correlation between proxy loss and perplexity. Shows convincing analysis of trade-offs between accuracy, overhead, and speed.

**Weaknesses:**

1. Although the proxy loss correlates with perplexity, a stronger justification (e.g., empirical correlation coefficients across models) would improve interpretability.

**Questions:**

1. How does the performance of ARMOR change with smaller calibration datasets or fewer proxy-loss iterations (e.g., 5k vs 20k)? Is the method robust to low-data calibration settings?
2. Could the block-diagonal wrappers be fine-tuned jointly with the model in a lightweight manner (e.g., LoRA-style), further improving performance?
3. Have you explored extending the framework to N:M patterns beyond 2:4? Since the design is general, it may adapt naturally.

---

> ### Author Response · Authors · 2025-11-21
>
> We thank Reviewer 4fMQ for their strong endorsement and for highlighting ARMOR's elegance, hardware friendliness, and theoretical rigor. We address the request for correlation analysis and robustness checks below.
>
> **W1: Proxy Loss Correlation With Perplexity**
>
> We appreciate the suggestion to quantify this relationship. To address this, we logged the global average proxy loss and perplexity (Wikitext-2 and C4) across optimization steps for Llama-2-7B, 13B, and Llama-3-8B. We then computed the Pearson correlation coefficient between the proxy loss sequence and the perplexity sequence.
>
> |  | Wikitext-2 | C4 |
> | :---: | :---: | :---: |
> | Llama-2-7B | 0.991 | 0.990 |
> | Llama-2-13B | 0.991 | 0.991 |
> | Llama-3-8B | 0.994 | 0.995 |
>
> The correlation is consistently $\geq$ 0.99, empirically confirming that the NoWag proxy loss is a faithful computationally tractable surrogate for model performance in the ARMOR framework.
>
> **Q1: Robustness to Calibration Data and Iterations**
>
> * **Iterations:** Performance improvement follows a log-log linear trend. For Llama-2-7B, at only 5k iterations (25% of the iterations used for the main results) ARMOR already recovers **87%** of the total perplexity gain. Linear regression on the log-log curve yields R^2 $\geq$ 0.96 across 3 models (Llama-2-7B/13B, Llama-3-8B) examined.
>
> *   **Calibration Data:** ARMOR is highly data-efficient. Reducing calibration data used to compress Llama-2-7B from 128 samples (default) to just **16 samples** results in negligible performance degradation (\<1% difference in), as shown below:
>
> |  | Tokens | Wikitext-2 | C4 |
> | :---: | :---: | :---: | :---: |
> | 16 Samples | 65.54K | 7.61 | 9.86 |
> | 32 Samples | 131.07K | 7.66 | 9.86 |
> | 64 Samples | 262.14K | 7.63 | 9.86 |
> | 128 Samples (Default) | 524.29K | 7.66 | 9.93 |
>
> For this experiment, a block size of 128 was used, and optimization was run for 5000 to expedite runtime.
>
> **Q2: Joint Fine-tuning of Wrappers (LoRA-style)**
>
> We agree that ARMOR's differentiable wrappers ($A$,$B$) make it uniquely suitable for LoRA-style fine-tuning compared to discrete mask-search methods. This opens a promising new direction for 'healing' pruned models that we plan to explore in a future work.
>
> **Q3: Extension to General N:M**
>
> Yes, ARMOR generalizes to any N:M pattern naturally. As a proof of concept (detailed in our response to G27K), we tested 4:8, 5:8, and 6:8 sparsity on Llama-2 7B/13B. In all cases, ARMOR consistently outperformed the NoWag-P baseline. For example, on Llama-2-7B (4:8 sparsity), ARMOR reduced Wikitext-2 perplexity from 8.04 to 6.74.

---

> > ### Comment · Reviewer_4fMQ · 2025-11-22
> >
> > Thank the authors for the new results. My concerns have been addressed. I will keep my score.

---

### Official Review · Reviewer_HqUF · 2025-11-01

**Soundness:** 3
**Presentation:** 3
**Contribution:** 3
**Rating:** 6
**Confidence:** 3

**Summary:**

The paper proposes **ARMOR**, a one-shot, *semi-structured* pruning method for N:M (notably **2:4**) that re-parameterizes each weight matrix as
\[
\hat W \;=\; A \; (W' \odot M) \; B,
\]
where \(M\) is a hardware-friendly **2:4** mask, and \(A,B\) are block-diagonal wrappers that act as lightweight, learnable pre-/post-transformations to **correct pruning error. The optimization alternates between: (i) continuous updates of \(A,B,W'\) under a NoWag-style proxy loss, and (ii) a greedy sparse-core update that exhaustively selects the best 2-of-4 configuration per group via small least-squares solves. The authors prove monotone convergence of the proxy loss and initialize at **NoWag-P**, guaranteeing a proxy loss no worse than that baseline. Empirically, on Qwen/Llama families, ARMOR reports **consistent accuracy gains over state-of-the-art **2:4** methods (Wanda / SparseGPT / NoWag-P) and provides inference measurements showing it largely retains 2:4 speed/memory benefits despite wrapper overhead.

**Strengths:**

- **Theoretically supported and rigorous.** Clear proxy-loss formulation, greedy 2:4 subproblem with closed-form LS updates, and a convergence theorem (monotone decrease of proxy loss; initialization at NoWag-P).
- **Strong accuracy vs. 2:4 baselines.** On Qwen/Llama, ARMOR consistently outperforms Wanda / SparseGPT / NoWag-P under 2:4 across PPL and downstream tasks; in some settings it narrows the dense–pruned gap by ~50%.
- **Insightful idea for future work.** Reframing pruning as adaptive factorization (sparse core + learnable wrappers) suggests a promising path for co-design—optimizing both representational form and hardware mapping, not only masks.

**Weaknesses:**

1. **Speedups are modest vs. expectations.** Reported generation speedups are **~1.041×–1.141×** on Qwen-2.5-14B/7B, and **batched MatVec** shows **1.57×** vs **1.86×** for plain 2:4—indicating nontrivial overheads or kernel gaps for wrappers. This is **far below** what a dense model with 50% fewer parameters could achieve, and below the **best-case 2:4** micro-kernel gains. A detailed **profile** of where time is spent (wrapper GEMMs, cache/bandwidth, kernel launch) is needed. :contentReference[oaicite:7]{index=7}
2. **Performance degration is significant compared to dense model and far below equal-parameter dense counterparts.** For example, ARMOR-pruned Qwen-2.5-14B can underperform Qwen-2.5-7B on some tasks; the absolute accuracy gap to dense remains notable in places, limiting immediate deployability. Consider exploring targeted post-training (e.g., LoRA or small SFT) to recover quality while preserving efficiency.
3. **Efficiency gap vs. plain 2:4 in microbenches.** The 1.57× batched MatVec vs 1.86× for 2:4 suggests wrapper-cost or fusion inefficiency. Without kernel-level fusion/tiling tailored to (block-diag × 2:4 × block-diag), ARMOR may remain below the Pareto frontier on pure efficiency. More benchmarks are needed to claim a better performance–efficiency trade-off overall.
4. **Tied strongly to NoWag-P and 2:4.** While the method is framed generally, in practice it heavily relies on the NoWag proxy + NoWag-P init and target **2:4**. Please clarify **algorithmic novelty** beyond NoWag-P, and quantify how much of the gain derives from more effective parameterization rather than more effective parameter count.
5. **Limited exploration.** It remains unclear whether combining ARMOR with structured pruning (e.g., head/column removal) and quantization (e.g., AWQ/GPTQ) yields **practical** end-to-end speedups while keeping quality. A hybrid study would strengthen deployability claims.

**Questions:**

1. **Where exactly is the runtime gap vs. 2:4?** Can wrapper fusion (e.g., pre-/post-accumulation, tiling into the 2:4 kernel) close the 1.57× vs 1.86× gap?
2. **Post-training recovery.** Have you tried small **LoRA/SFT** passes after ARMOR to close the accuracy gap without losing efficiency?

---

> ### Author Response · Authors · 2025-11-21
>
> We thank Reviewer HqUF for their time and for highlighting the theoretical rigor and strong accuracy of ARMOR. We address the specific weaknesses and questions below:
>
> **W1 & W3 & Q1: Inference Speedups & Kernel Optimizations** The reviewer notes that our 1.57x batched MatVec speedup (vs 1.86x for naive 2:4) indicates overhead. This is true and expected, **ARMOR trades a small fraction of theoretical throughput for significant gains in model quality (reducing the perplexity gap by ∼50% vs existing 2:4 one-shot pruning algorithms) and reduces the memory requirements** **needed to perform inference.**
>
> The overhead comes from the block-diagonal pre/post-multiplications. As noted in the paper, this scales linearly with O((dout​+din​)dblock​). For the specific Qwen-2.5 14B gate projection layer (din \= 5120, dout \= 13824\) that MatVec experiments were performed on, the block diagonal wrappers (d=128) add a 3.4% flop overhead. Our pytorch only implementation of ARMOR already archives 84% of the theoretical maximum speedup (1.57/1.87)  By comparison, the highly optimized plain 2:4 kernel (1.86×) achieves ≈93% of its theoretical limit. We agree that a custom kernel could close this gap even further, however, even with the current pytorchimplementation, ARMOR provides actual speedups (1.14x end-to-end) while SOTA naive methods often fail to preserve performance at this sparsity level.
>
> **W2 & Q2: Performance degradation and Post-Training Recovery (SFT/LoRA):** We believe the reviewer is mistaken in direct comparisons to the dense models. ARMOR is a one-shot pruning method. The standard evaluation protocol for this class of algorithms (SparseGPT, Wanda, etc.) is to evaluate the performance immediately after pruning to specifically isolate the sensitivity of the pruning metric. As a result, we expect our performance to be lower than that of comparable dense models, since no training is done. Furthermore, while ARMOR far below equal-parameter dense counterparts at some benchmarks, for others **ARMOR is close or even exceeds roughly comparable dense models solely by one-shot pruning**, for example MMLU, BBH, GPQA for Qwen-2.5 32B/72B.
>
> While we agree that fine-tuning (SFT) would likely improve results further, it introduces confounding variables (training data, hyperparameters) that obscure the contribution of the factorization itself. We believe combining ARMOR with LoRA is a promising direction for future work, but strictly outside the scope of evaluating the pruning algorithm's efficacy.
>
> **W4 Tied strongly to NoWag-P and 2:4.** We strongly disagree with the reviewer’s characterization that ARMOR is strongly tied to NoWag-P and 2:4 semi-structured sparsity. Below we address each characterization in depth:
>
> **Tied to NoWag:** To clarify, **the ARMOR optimization algorithm is proxy loss agnostic**. As long as the chosen proxy loss is differentiable and can be decomposed into element wise subproblems, ARMOR can work. As a demonstration, we compared ARMOR pruned Llama 2 7B with the NoWag proxy loss vs ARMOR pruned Llama 2 7B with a Wanda pruning metric:
>
> |  | Wikitext-2 PPL ($\\downarrow$) | C4 PPL ($\\downarrow$) |
> | :---- | :---- | :---- |
> | ARMOR+Wanda | 7.65 | **9.88** |
> | ARMOR+NoWag | **7.59** | **9.88** |
>
> Block size 128 and 5000 iterations were performed for both. The performance gap is negligible (\<1%), demonstrating that **ARMOR's gains come from the factorization architecture**, not the specific proxy loss. We utilize NoWag simply because it is slightly more performant.
>
> **Tied to 2:4** While the experiments we presented were solely for 2:4 sparsity, this was simply because this pattern is supported natively by Nvidia GPUs. The ARMOR algorithm works with any N:M pattern out of the box, please see the experiments performed in our response to Reviewer G27K for 4:8, 5:8, and 6:8. And with a slight modification, ARMOR can also work for unstructured sparsity, please see our response to reviewer Ze6L.
>
> **W5: Extensions to Structured Sparsity and Quantization:**
> We agree with the reviewer that incorporating structured sparsity and quantization with ARMOR is an interesting idea. However such directions are orthogonal, ARMOR provides a better starting representation $\\hat{W}$ or these techniques. A full combinatorial study of ARMOR \+ Quantization \+ Structured Pruning vs baselines would be beyond the computational resources we have available.

---

### Author Response · Authors · 2025-12-02

We would like to extend our most sincere gratitude to our reviewers, initial and re-assigned AC in these unprecedented times.

We are particularly encouraged that all reviewers reached a consensus on the significance, theoretical rigor, and novelty of our adaptive factorization approach. We also appreciate that the reviewers update their scores to (6, 8, 8, 8\) following our responses, prior to the reversion due to the OpenReview incident. Based on their constructive feedback we have made several revisions and improvements to the manuscript. Below we have listed the key ones:

**Key Revisions & Additions:**

* **Generalization to N:M and Unstructured Sparsity (Section 4.5 and Table 6):** We added results for 4:8, 5:8, and 6:8 semistructured sparsity , and 50% unstructured sparsity, demonstrating that ARMOR exhibits consistent gains over baselines across all patterns (Reviewer 4fMQ, Ze6L, G27K).
* **Improved Literature Review and Comparisons with Learnable Baselines (Section 1.1, 4.3 and Table 5):** We have revised the literature review (Section 1.1) to incorporate the works Reviewers Ze6L and G27K have mentioned. Additionally we have included comparisons with all relevant works, including contemporary works, that have reproducible code (Section 4.3 and Table 5).
* **Rewritten Inference Efficiency Analysis (Section 4.3):** We expanded the discussion on inference speedups, clarifying the specific trade-offs between theoretical throughput (wrappers) and accuracy gains (Reviewer HqUF, G27K).
* **Extension to MoE Models (Appendix F):** We validated ARMOR on Qwen3-30B-A3B MoE, showing it outperforms baselines and works out-of-the-box for Mixture-of-Experts architectures (Reviewer Ze6L).

We believe these additions significantly strengthen the paper's empirical breadth and theoretical clarity.

---

### Meta-Review · Area_Chair_k2PF · 2026-01-05

**Summary:**

The paper proposes a post-training semi-structured pruning method named ARMOR that factorizes weight matrices into a sparse core (e.g., 2:4) wrapped by block-diagonal matrices to correct pruning errors. Reviewers unanimously recognized the novelty, theoretical rigor, and practical value of the adaptive matrix factorization approach. During the rebuttal, the authors successfully addressed concerns regarding generality by extending the method to N:M patterns, unstructured pruning, and MoE models, and provided comprehensive comparisons against state-of-the-art learnable pruning baselines. The reviewers reach consensus that ARMOR establishes a superior trade-off between compression and accuracy compared to existing methods like Wanda and SparseGPT.

**Reviewer Concerns:**

**Addressed reviews:**
1. **Generality & Extensions (Reviewer Ze6L, G27K, 4fMQ)**: Reviewers initially questioned if the method was limited to 2:4 sparsity or dense models. The authors provided new results for 4:8, 5:8, 6:8 patterns, 50% unstructured pruning, and Qwen3-MoE models, demonstrating consistent gains.
2. **Baselines & Novelty (Reviewer G27K, Ze6L)**: Concerns about limited comparison with learnable baselines (e.g., RotPruner, DenoiseRotator) were addressed. The authors added comparisons showing ARMOR outperforms these methods and clarified the distinction from retraining-based methods like MaskLLM.
3. **Inference Efficiency (Reviewer HqUF, G27K)**: Reviewers noted the gap between theoretical and actual speedups. The authors clarified the trade-off, explaining that the slight throughput reduction (due to wrapper overhead) allows for significantly higher accuracy, achieving ~84% of the theoretical maximum speedup.
4. **Proxy Loss Correlation (Reviewer 4fMQ)**: The request for justification of the proxy loss was met with a >0.99 Pearson correlation with perplexity.

**Outstanding concerns** (Partially Resolved):
1. Reviewer HqUF retained a concern about the lack of exploration into combining ARMOR with quantization or SFT, and felt the speedup was still "modest". The authors argued that these are orthogonal directions or future work. The AC found the rebuttal promising though Reviewer HqUF did not provide follow-up discussions.

**Reviewer Scores:**

- Reviewer HqUF: 6 - Did not explicitly raise score, likely maintained 6.
- Reviewer 4fMQ: 8 - Maintained score after concerns were addressed.
- Reviewer Ze6L: 6 -> 8 - Raised score after MoE and unstructured pruning experiments were added.
- Reviewer G27K: 6 -> 8 - Raised score after additional baseline comparisons and efficiency analysis.

---

### Decision · Program_Chairs · 2026-01-26

Accept (Poster)